# Fatigue Analysis of Threaded Components with Cd and Zn-Ni Anticorrosive Coatings

**Jefferson Rodrigo Marcelino dos Santos** *[ID], **Martin Ferreira Fernandes** [ID], **Verônica Mara de Oliveira Velloso** and **Herman Jacobus Cornelis Voorwald**

Fatigue and Aeronautical Materials Research Group, Department of Materials and Technology, School of Engineering, Sao Paulo State University (UNESP), Guaratingueta 12516-410, SP, Brazil; martin.fernandes@unesp.br (M.F.F.); veronicamcaoliveira@gmail.com (V.M.d.O.V.); h.voorwald@unesp.br (H.J.C.V.)
* Correspondence: jffrodrigo@yahoo.com.br

**Abstract:** The influence of the electrodeposition of cadmium and zinc-nickel and the stress concentration effect on the fatigue behavior of AISI 4140 steel threaded components were studied. Axial fatigue tests at room temperature with a stress ratio of R = 0.1 were performed using standard and threaded specimens with and without nut interface under base material, cadmium, and zinc-nickel-coated conditions. Finite element analysis (FEA) was used, considering both elastic and elastoplastic models, to quantify the stress distribution and strain for threaded specimens with and without a nut interface. The numeric results were correlated to the experimental fatigue data of threaded components with and without the nut interface, to allow the oil & gas companies to extrapolate the results for different thread dimensions, since the experimental tests are not feasible to be performed for all thread interfaces. Scanning electron microscopy (SEM) was used to analyze the fracture surfaces. The stress concentration factor had a greater influence on the fatigue performance of threaded components than the effect of the Cd and Zn-Ni coatings. The fatigue life of studs reduced by about 58% with the nut/stud interface, compared to threaded components without nuts. The elastoplastic FEA results showed that studs with a stud/nut interface had higher stress values than the threaded specimens without a nut interface. The FEA results showed that the cracks nucleated at the regions with higher strain, absorbed energy, and stress concentration. The substitution of Cd for a Zn-Ni coating was feasible regarding the fatigue strength for threaded and smooth components.

**Keywords:** fatigue; AISI 4140 steel; finite element method; stud; FEA

## 1. Introduction

Oil and gas are extracted from the natural environment under high pressures. The transport from the well to the platform requires equipment like a wellhead, "Christmas tree", PLET, PLEM, manifolds and pipes, which need to withstand internal pressure up to 20,000 psi. Any failure could be catastrophic for the environment, people, and oil companies. To withstand the load due to pressure and avoid leakage, the studs are preloaded by torque application on nuts to create a tensile stress value of around 362 MPa, which could reach 600 MPa during working or testing conditions. Finite element analysis (FEA) aims to show the stress profile in those components, and the investigation of fatigue behavior is essential to prevent catastrophic failures.

For the equipment used in seawater, oxidation is also an issue. In order to prevent the corrosion of studs, an electrodeposited coating of cadmium or zinc-nickel is essential for oil and gas applications, to improve wear and corrosion properties [1,2]. However, coatings influence the fatigue behavior of components [3,4]. Although surface treatments may have some positive impacts on the fatigue strength, such as high mechanical strength, compressive residual stresses, and good adhesion, the coatings frequently reduce the fatigue life of components through the effect on the fatigue crack initiation period [5,6].

An electroplated chromium coating significantly affects fatigue behavior due to tensile residual stresses and microcracks [7]. Cadmium electrodeposition may reduce the fatigue strength due to hydrogen embrittlement [8]. Therefore, the fatigue lifetime of Cd-coated components could decrease compared to the base metal, due to cracks at the coated surface and due to the hydrogen embrittlement induced by the electrodeposition process. For studs used for testing purposes, it is essential to quantify the number of cycles under the worst conditions to avoid catastrophic failures.

Cadmium is frequently applied as an anticorrosive coating due to a higher negative corrosion potential than steel [9]. However, cadmium use is being restricted due to its carcinogenic and toxic nature and the risk it poses to the environment [9,10]. Several studies address acceptable alternatives to cadmium for different applications [11–13]. Zn-Ni alloy coatings are a potential alternative, with better corrosion protection for marine environments than pure Zn [14]. Previous studies reported that Zn-Ni coatings had better corrosion resistance [15], microhardness [3], adhesive wear [3], and corrosion resistance during wear [16] than cadmium coatings. The fatigue behavior of Cd- and Zn-Ni coated smooth specimens was investigated in a previous paper [17]. The work proved the feasibility with respect to the fatigue behavior of the replacement of electrodeposited Cd with a Zn-Ni alloy [17]. However, the study still needs further fatigue and FEA analysis to understand the fatigue behavior of studs applied in the oil and gas industries.

The finite element analysis (FEA) is a numeric technique to solve differential equations for several engineering problems, based on the discretization of a continuous geometry into a mesh of several finite elements. The equations that model these finite elements are assembled into a system of equations that models the entire problem. The FEA application covers a wide range of engineering and scientific problems [18–21]. Owolabi et al. studied the notch root radius effect on the probability of failure [18]. Chaves et al. correlated the stress gradient at the notch obtained through FEA to the fatigue behavior [19]. Gao studied through FEA the influence of the theoretical stress concentration factor (Kt) on the fatigue strength of notched steel specimens [20]. In the present paper, the FEA technique was applied to investigate the stress distribution in studs applied in oil and gas industries and correlate the numeric results to the fatigue behavior of threaded components with and without the nut interface. The FEA aims mainly to identify the critical crack nucleation points on threaded components. The coating effect was not addressed in simulations.

In this paper, axial fatigue tests were performed for AISI 4140 steel for smooth, threaded specimens and studs with and without nut interface under base material, cadmium, and zinc-nickel coating conditions. The fatigue tests were performed at room temperature with sinusoidal waveforms, a stress ratio of 0.1, and a frequency of 20Hz. Finite element analysis was used to describe the stress distribution across the section of the studs to make a comparison between threaded specimen configurations with and without nut interface.

## 2. Materials and Methods

### 2.1. Material and Coatings

The material used in this work is AISI 4140 steel produced by an electric arc furnace with the chemical composition shown in Table 1. The material was received in a hot-rolled mill condition in the form of rods. The heat treatment of the manufactured specimens was composed of homogenization at 900 °C for 3 h, followed by quenching in oil at 81 °C, and, finally, double tempering treatment (620 °C for 3 h, cooling in calm air), reaching a yield strength of 860 MPa, ultimate tensile stress of 978 MPa, and elasticity modulus of 202 GPa. The mechanical properties were obtained through tensile tests of 3 smooth specimens according to ASTM E8/E8M. The microstructure is tempered martensite. For the cadmium coated specimens, 13 μm of Cd was applied, and dehydrogenation treatment was performed (200 °C for 8 h). For the zinc-nickel coated specimens, the coating thickness was in the range of (5–10) μm, and dehydrogenation treatment was also performed (200 °C for 8 h).

**Table 1.** Chemical composition of AISI 4140 steel (wt%).

| C | Mn | P | S | Si | Cr | Mo | V | B | Fe |
|---|---|---|---|---|---|---|---|---|---|
| 0.38 | 0.75/1.00 | <0.035 | <0.04 | 0.15/0.35 | 0.8/1.10 | 0.15/0.25 | <0.05 | 0.0005 | Base |

An optical microscope Nikon Epiphot 200 (UNESP, Guaratingueta, Brazil) was used to characterize the microstructure, and the material was etched using the following solutions: (i) 2% Nital, applied by immersion with an exposition time of 20 s, (ii) with 10% sodium metabisulfite for 20 s, and (iii) with LePera reagent (1% sodium metabisulfite and 4% picric acid solutions in the proportion 1:1) for 25 s. Scanning electron microscopy (SEM) analysis with LEO model 1450-VP equipment (UNESP, Guaratingueta, Brazil) was used to characterize the fracture surfaces obtained after fatigue tests.

*2.2. Experimental Fatigue Tests*

The geometry of specimens experimentally and numerically analyzed in this work is displayed in Figure 1. The specimen configurations tested are (a) standard specimens according to ASTM E466, (b) 1/4"–20 TPI threaded specimens, and (c) 1/4"–20 TPI studs. The thread of studs was manufactured by the rolling process that generates compressive stress on the surface, increasing the fatigue life compared to machined threads. The axial fatigue tests were performed for the geometries shown in Figure 1. The fatigue tests were carried out with sinusoidal waveforms, stress ratio R = 0.1, and frequency f = 20 Hz at room temperature using an Instron 8801 testing machine (UNESP, Guaratingueta, Brazil) and hydraulic wedge gripping system (UNESP, Guaratingueta, Brazil). It is noteworthy that no torque was applied between the nut and sleeve for the testing using the nut interface. All tests were performed in the air environment. Despite the testing environment not reflecting the real subsea application, testing in the air is quite important to obtain faster results. For extrapolation, taking into account the seawater and cathodic protection, the oil companies should use international standards [22].

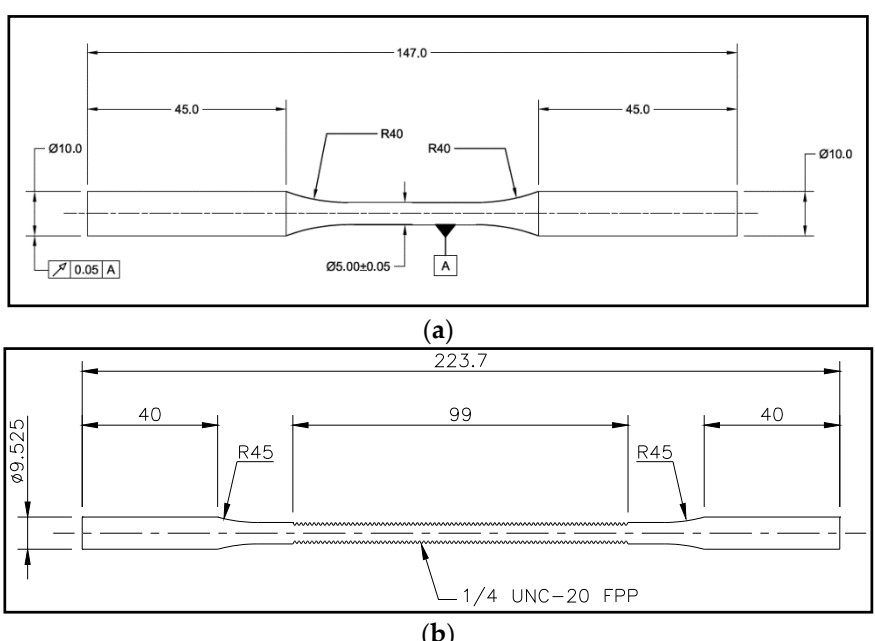

(a)

(b)

**Figure 1.** *Cont.*

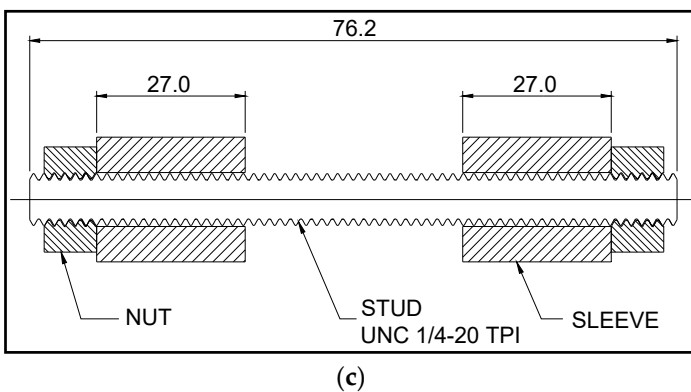

(**c**)

**Figure 1.** Detail of specimens used for testing. (**a**) Standard specimen according to ASTM E466, (**b**) threaded specimen 1/4"–20 TPI (dimensions in millimeter, except thread), (**c**) stud 1/4"–20 TPI specimen (with nut interface-dimensions in millimeters, except thread).

*2.3. Finite Element Analysis*

This paper aims to study the application of steel studs in petrochemical companies, where both the coating and the stress concentration effect play a significant role in fatigue performance. The finite element analysis in the present work aims to clarify the effect of stress concentration in studs and threaded specimens, and the coating effect was not considered in simulations. The finite element analysis (FEA) was used to investigate the stress distribution of a particular configuration of threaded components and correlate it with the experimental results. Based on this comparison, the behavior of other geometries can be predicted, and significant experimental work can be saved.

The simulations used axisymmetric models to quantify the stress level for threaded specimens during application. For the specimens without nut interface, the model was made of 15,791 nodes and 14,690 elements of type CAX4R, refining the threaded regions highlighted in Figure 2a. For the specimens with nut interface, the model was made of 17,243 nodes and 16,211 elements of type CAX4R, as shown in Figure 2b. The friction was considered to be 0.13 at contact pairs, reflecting the real system where no grease or oil was used.

Elastic and elastoplastic finite element analyses were performed for both threaded and nut base material specimens, to investigate the stress and strain distribution across the section under the highest stress level and the strain energy for elastoplastic conditions. For threaded specimens, the stress was measured relative to the coordinate system shown in Figure 3a,b for specimens without and with a nut interface, respectively, in the "x" direction.

The boundary conditions used for FEA, shown in Figure 4, represent the physical system, where one side was fixed, and the other receives the load, simulating the geometry attached to the wedge grip system.

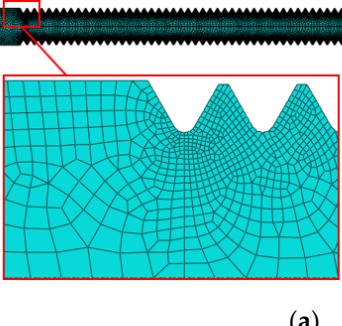

(**a**)

**Figure 2.** *Cont.*

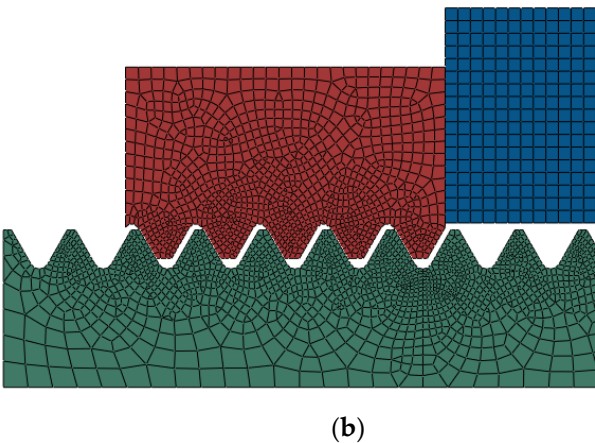

(**b**)

**Figure 2.** Detail of finite element mesh for (**a**) the stud without nut interface, and (**b**) interface stud/nut/sleeve.

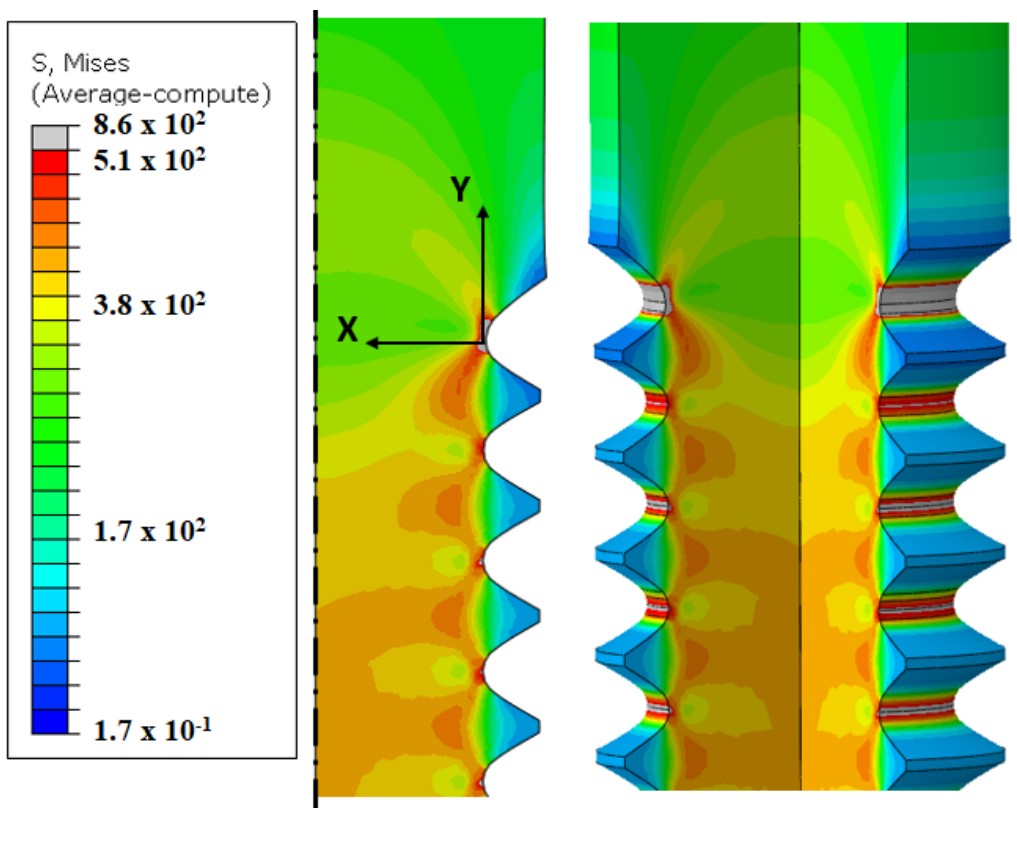

(**a**)

**Figure 3.** *Cont.*

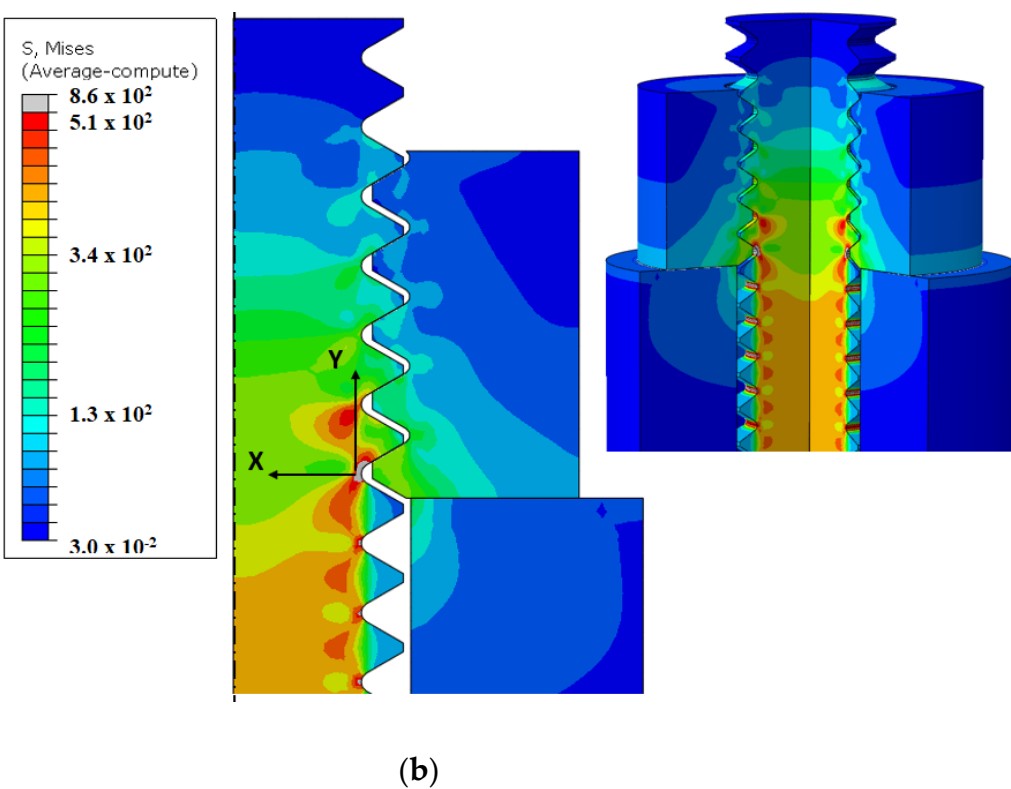

**(b)**

**Figure 3.** Local coordinate system to quantify stress from the root of the thread (**a**) without nut interface and (**b**) with nut interface.

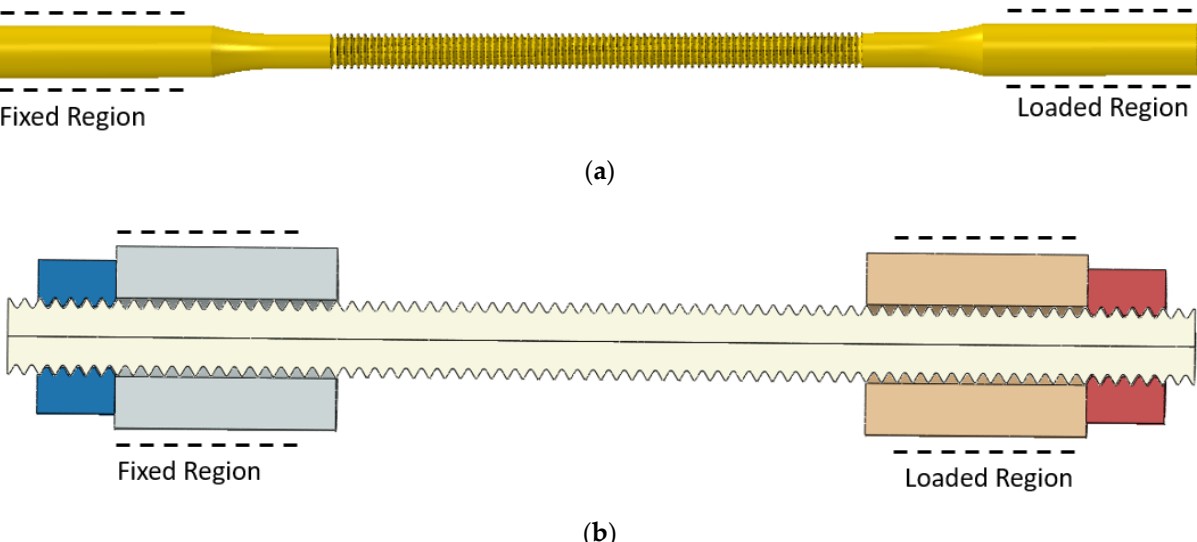

**Figure 4.** Regions attached to the wedge grip system (**a**) threaded specimen (**b**) stud specimen.

## 3. Results and Discussion

### 3.1. Microstructure and Coating Characterization

Figure 5a shows that the microstructure revealed after 2% Nital chemical etching is tempered martensite. The as-quenched martensite phase was formed after the diffusionless transformation of austenite from 900 °C to 81 °C, and the subsequent double tempering of the material at a temperature of 620 °C allowed the formation of tempered martensite, composed of the stable ferrite (white color) and cementite (dark color) phases. The tempered martensite microstructure is important for fatigue behavior due to an increased fracture

toughness compared to the as-quenched martensite microstructure. During the first tempering, carbide precipitation occurred in the retained austenite, reducing the carbon content in this phase, which increased the temperature of the martensitic transformation zone. In the second one, the new martensite phase was tempered. Nevertheless, it was identified retained austenite phase after 10% metabisulfite etching, which is the polygonal white shape forms outlined in red (Figure 5b). Due to the double tempering heat treatment, there is a low amount of austenite phase.

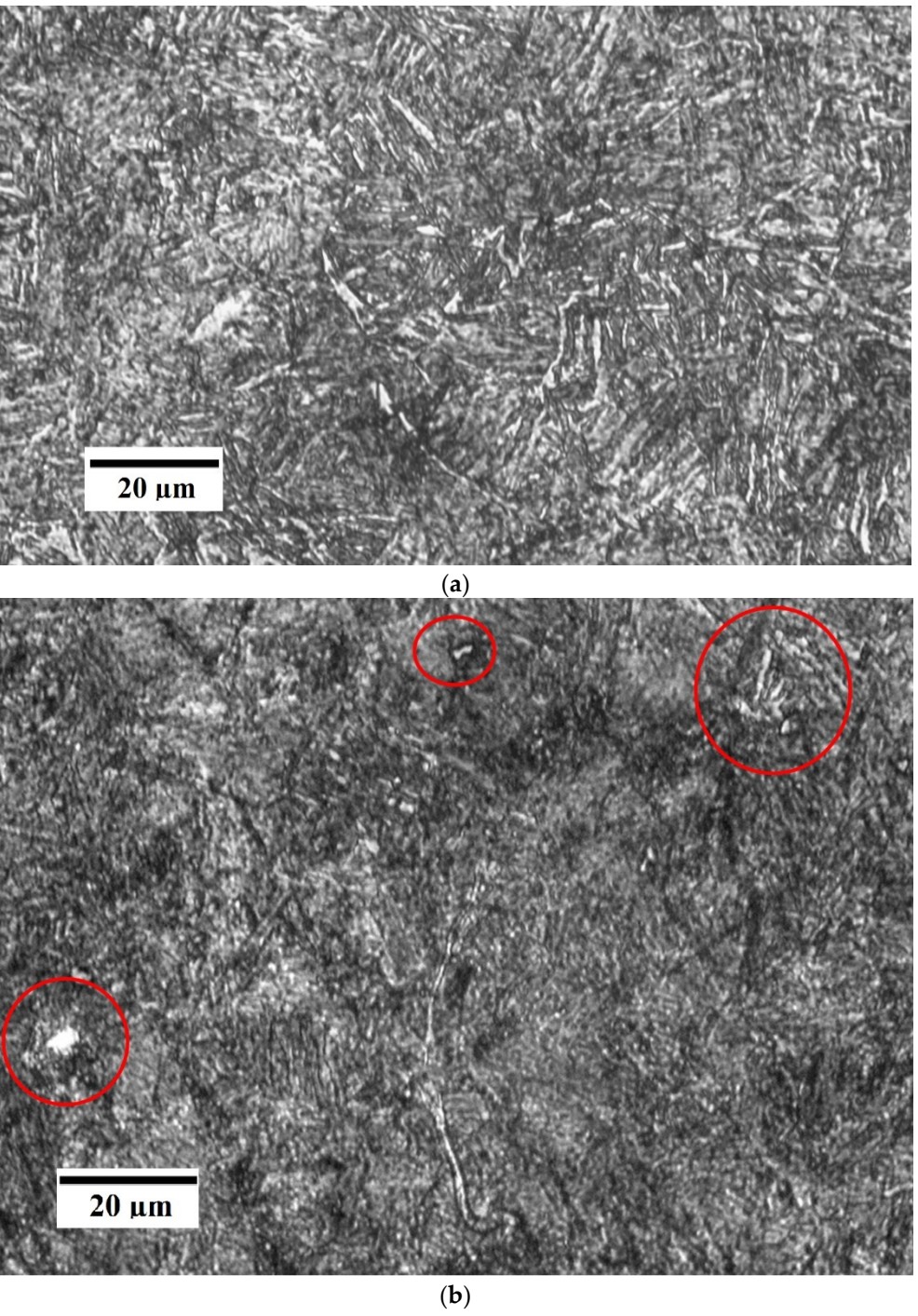

(a)

(b)

**Figure 5.** *Cont.*

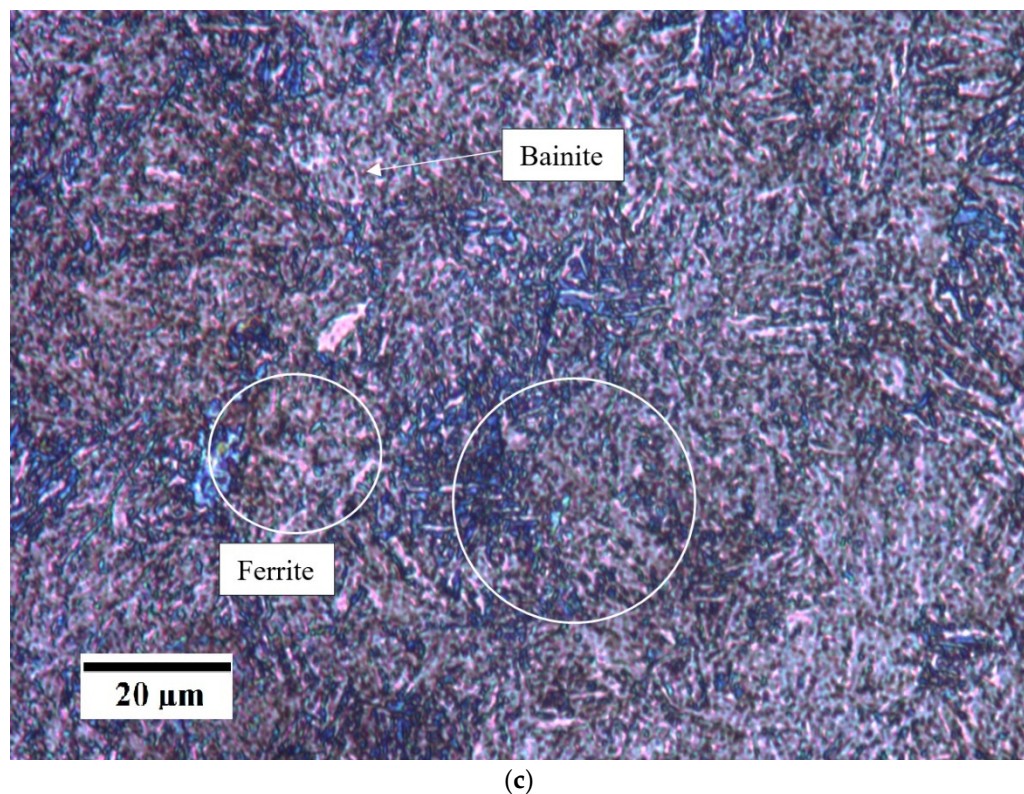

(**c**)

**Figure 5.** Microstructure after (**a**) 2% Nital etching (200×), (**b**) 10% sodium metabisulfite etching (500×), (**c**) reagent LePera etching–OM (500×).

The reagent LePera chemical etching revealed different phases: the lighter color phases are of tempered martensite and retained austenite, the blue phase is ferrite, and the brown phase is bainite (Figure 5c).

Figure 6 shows OM images of the Cd and Zn-Ni electroplated coatings at a magnification of 500×. It is possible to verify that the coating layers were uniform, with good homogeneity and strong adhesion to the substrate.

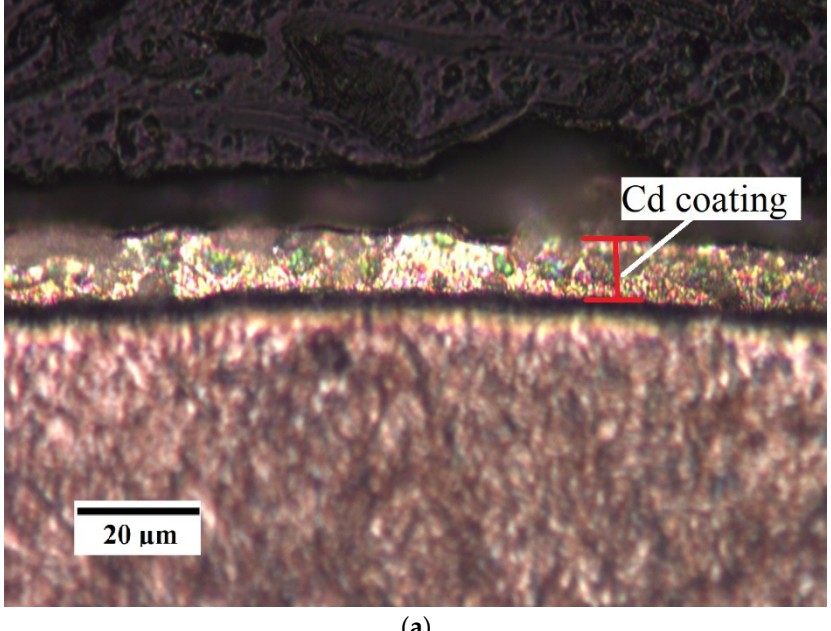

(**a**)

**Figure 6.** *Cont.*

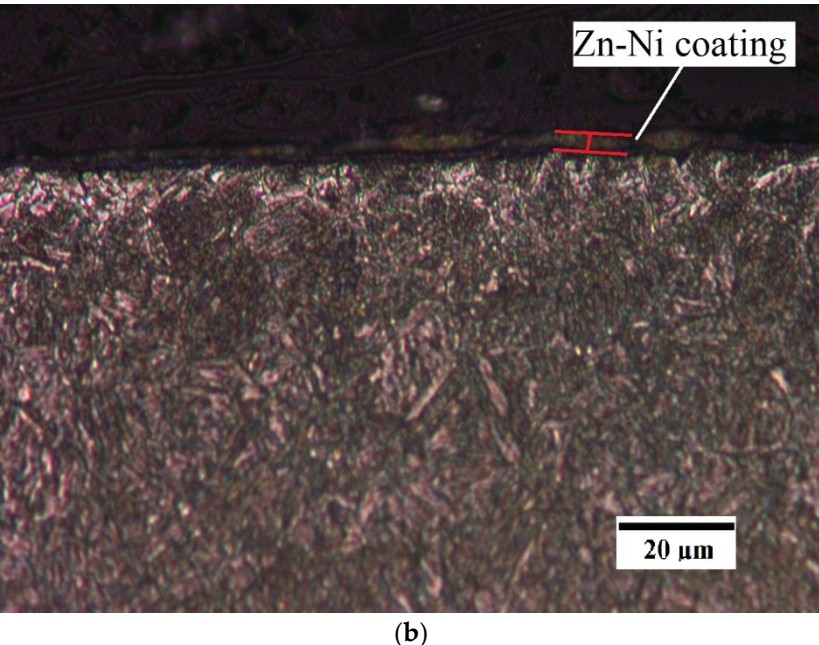

**(b)**

**Figure 6.** Optical microscopy images of (**a**) Cd and (**b**) Zn-Ni electroplated coatings (500×).

### 3.2. Fatigue Data

The fatigue tests at room temperature and air environment were performed for the base material, Cd and Zn-Ni alloy coated AISI 4140 steel. The specimen configurations were standard specimens according to ASTM E466 (smooth specimens), threaded specimens, and studs. First, the fatigue behavior of smooth specimens with coatings was investigated and compared to the base material. Second, the fatigue tests were performed for threaded specimens with coatings and the base material. Third, the fatigue behavior of studs was investigated and compared to the threaded specimen base material. The maximum stress values for smooth and threaded specimens were obtained from the minimum diameter for each specimen. Care should be taken herein because the computed stress could be slightly lower since the thread is not axisymmetric.

The fatigue behavior of Cd and Zn-Ni smooth specimens was investigated in a previous work [17]. The experimental results for coated smooth specimens are relevant to this work to identify the influence of the coatings on the fatigue behavior without the superposition of the stress concentration effect in the case of threaded specimens and studs. Figure 7 shows the S-N curves for standard specimens (Figure 1a) under base material, cadmium, and Zn-Ni conditions. The fatigue life increased for both Cd and Zn-Ni conditions. The explanation for this behavior is the residual stress field generated near the surface due to the electrodeposition process and the effect of dehydrogenation heat treatment on the residual stress field [8,17]. The difference between the fatigue life of the two coated conditions could be explained by the difference in residual stress levels and coating thicknesses. However, the statistical dispersion of the results contributes to a nearly similar statistical behavior of the fatigue data of the base material, and Cd coated conditions at a stress level of 750 MPa. It is noteworthy that the shape parameters of the Weibull distribution for all stress levels tested with at least three specimens were always higher than 1, which indicates a high reliability of the results and homogeneity of the material.

Figure 8 shows S-N curves for threaded specimens without a nut interface (Figure 1b) for base material, cadmium, and Zn-Ni coating conditions. There was no relevant influence of the coating process on the fatigue behavior, due to the strong influence of the threaded geometry. Furthermore, it is also possible to assume that the dehydrogenation treatment was effective since the fatigue life was similar for both coated and uncoated specimens. The results showed that both Cd and Zn-Ni coatings do not significantly influence the

fatigue behavior of the AISI 4140 steel threaded specimens. Therefore, the coatings could be applied in threaded components without detrimental effects on fatigue performance.

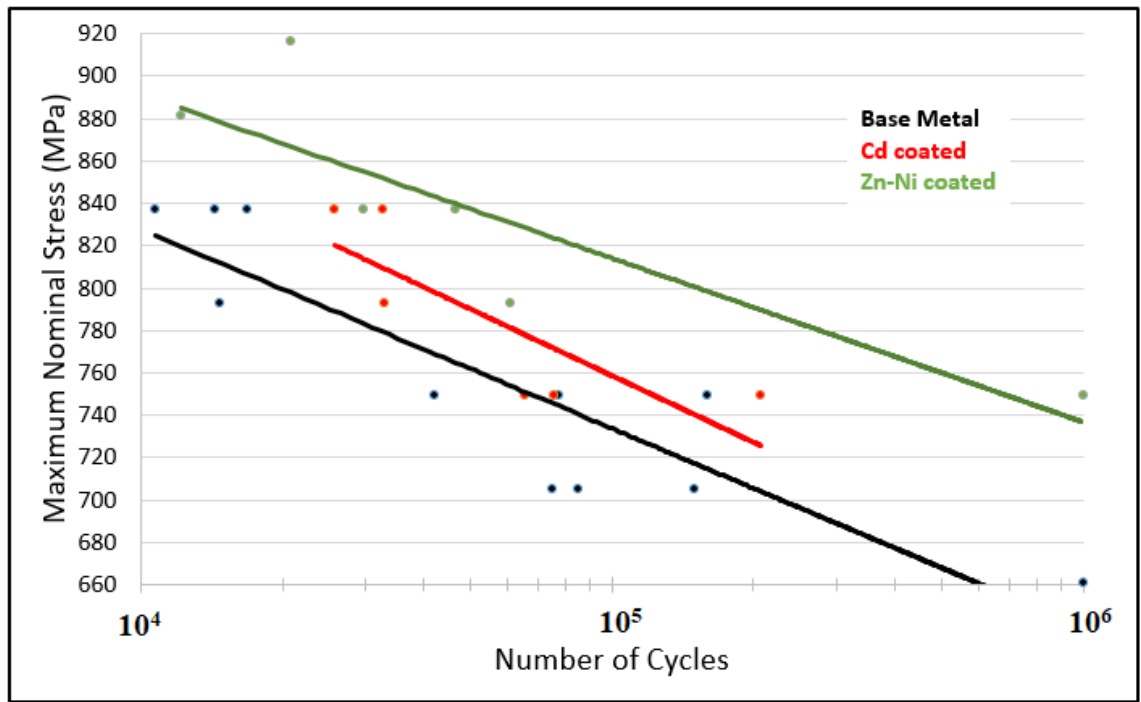

**Figure 7.** Fatigue life, standard specimens (ASTM E466) [17].

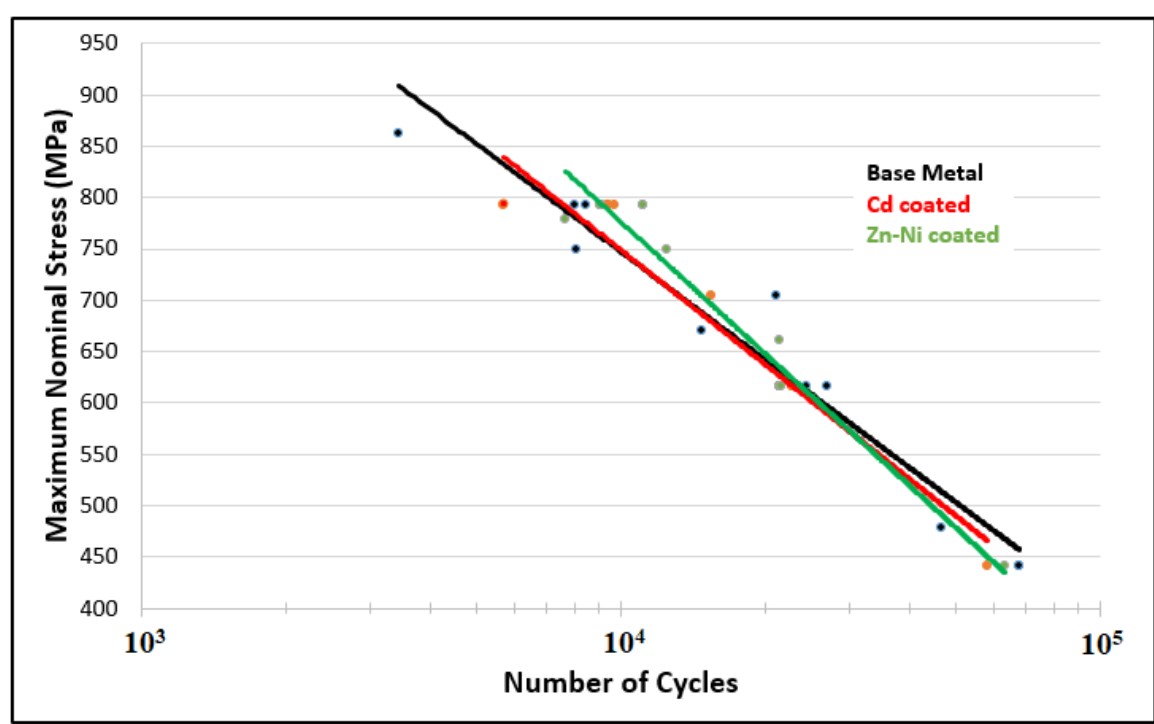

**Figure 8.** Fatigue life, threaded specimens.

In order to investigate the fatigue performance of real studs applied in the oil and gas industry, fatigue tests were performed for the studs with nuts assembled (Figure 1c). Figure 9 shows the S-N curves for both threaded (Figure 1b) and stud specimens (Figure 1c) without coatings or coated with Cd or Zn-Ni; as we have identified, the coating effect is not

quite relevant for thread geometries. It was possible to realize that the fatigue life reduction was about 58% for the condition with the nuts assembled and, consequently, the presence of the nut/stud interface. Therefore, there was a high contribution to reducing fatigue life when there was a contact between male and female threads.

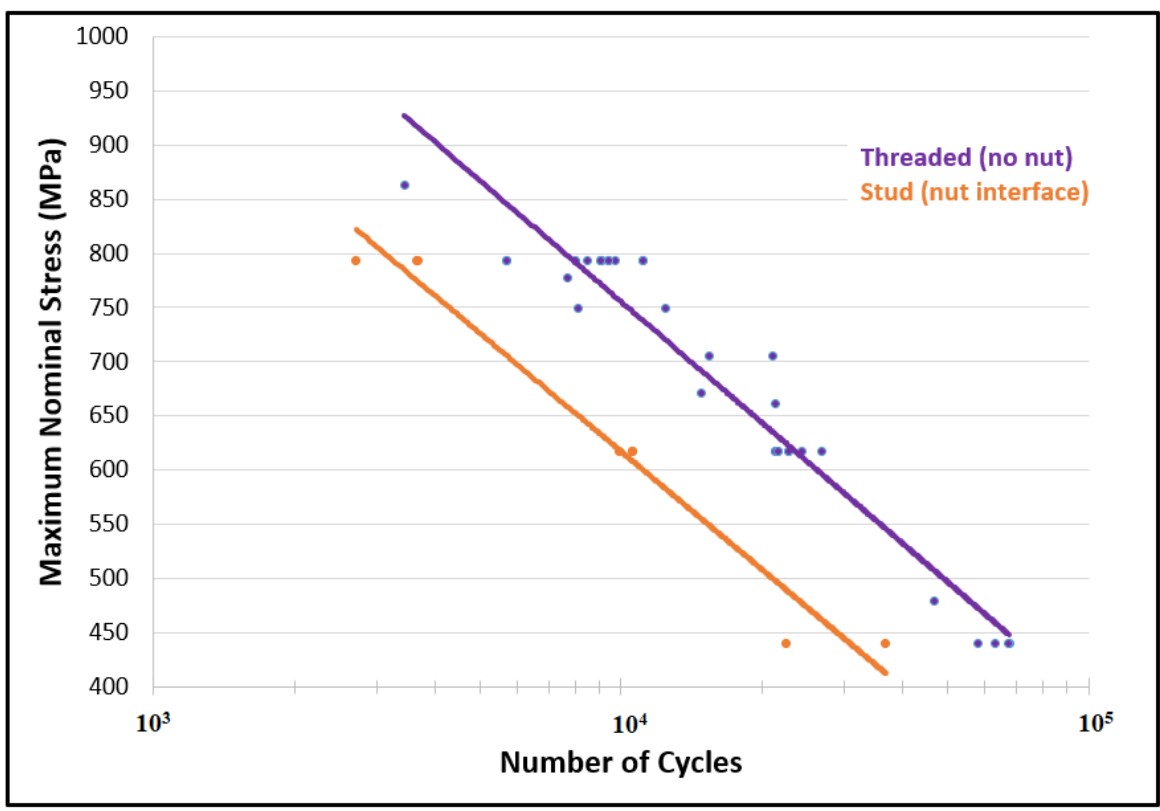

**Figure 9.** Fatigue life comparison (with and without nut interface).

### 3.3. Finite Element Analysis

3.3.1. Elastoplastic Analysis

　　The stress gradient is a valuable tool to understand the stress concentration effect and its influence on fatigue performance. Figure 10 displays the von Mises stress (MPa), obtained through an elastoplastic analysis for studs with nut interface at the situations where the maximum nominal stress values at the studs were 440.6 MPa, 616.8 MPa, and 793 MPa, which correspond to 50%, 70%, and 90% of the yield strength, respectively. Although most of the cross-sectional areas of the studs withstand the nominal stress values, there were points near the threads where the stress concentration effect resulted in higher stress values. The highest stress concentration regions explain the fatigue crack nucleation sites and the reduction in the fatigue life of the studs compared to the smooth base material.

　　Figure 10 shows in gray color the regions where the von Mises stress was higher than the nominal stress. It was possible to realize that all the surfaces of thread roots were under high stress due to the stress concentration factor. Close to the nut, the stress reached even higher values. The reason for this behavior is the bending and shear effects by the contact between male and female threads. The analysis that supports this conclusion is presented in Figure 11, where the tensile load between the last engaged threads (LET) is the same along the threads shown in Figure 10a. The ratio between the maximum and nominal stress values decreased as the maximum stress of the FEA increased, reaching 1.95, 1.41, and 1.11 for nominal stress levels of 440.6 MPa, 616.8 MPa, and 793 MPa, respectively. The reason for this behavior is mainly because stress relaxation is a characteristic of self-limiting secondary stress.

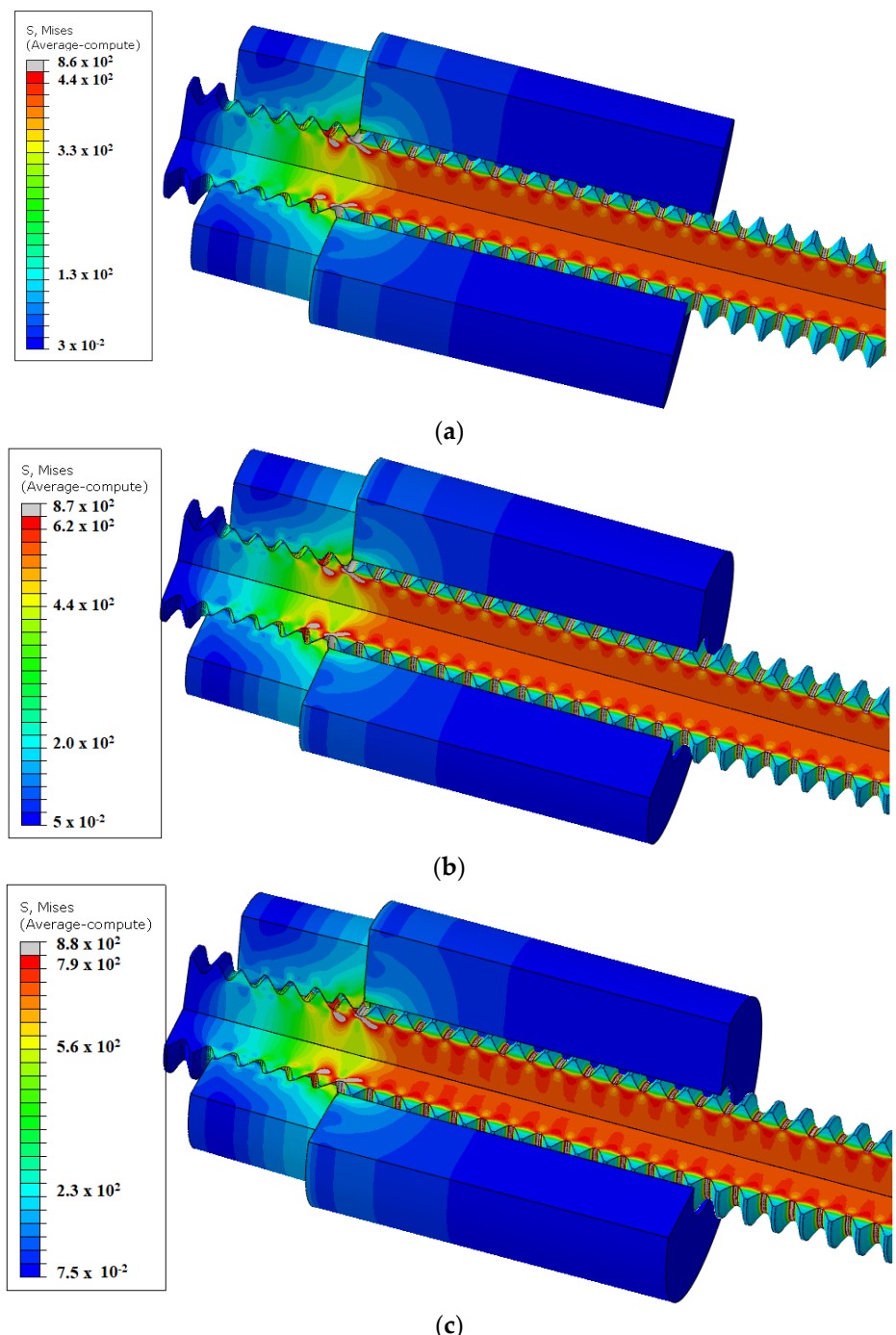

**Figure 10.** FEA for studs with nut interface. The von Mises stress (MPa) is shown for maximum nominal stress of (**a**) 440.6 MPa (50% of the yield strength), (**b**) 616.8 MPa (70% of the yield strength), and (**c**) 793 MPa (90% of the yield strength).

Figure 12 shows the strain (mm/mm) for studs with nut interface at the situations where the maximum nominal stress values at the studs were 50%, 70%, and 90% of the yield strength; it is possible to discern that even for nominal stress values of 50% and 70% of the yield, a small region at the bottom of the thread reached plasticity, due to the stress concentration. The regions with higher strain values, pointed by arrows, correspond to the regions with higher plastic deformation processes, which is also responsible for reducing the fatigue crack nucleation period. The plastic deformation accumulation increased with the increase of the applied stress, which is also related to a detrimental effect on the fatigue life of the components. The color regions in Figure 12 correspond to the plastic deformation

occurring at the root of the LET, and, therefore, the probability of nucleating a crack at this region is higher. Figure 12c shows the condition where the nominal stress combines 90% of the yield strength and the plastic deformation, resulting in stress relaxation.

Figure 13 shows the absorbed energy (J) for studs with nut interface for the situations where the maximum nominal stress values at the studs were 50%, 70%, and 90% of the yield strength. The maximum absorbed energy values were about 0.66, 1.3, and 2.2 J for the stress values of 440.6 MPa, 616.8 MPa, and 793 MPa, respectively. The higher absorbed energy values are responsible for the reduction of the fatigue life of studs.

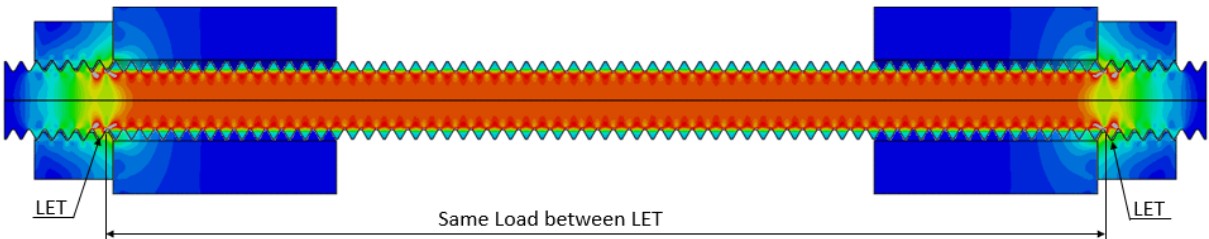

**Figure 11.** Detail of FEA for studs showing the same tensile load between the last engaged threads (LET).

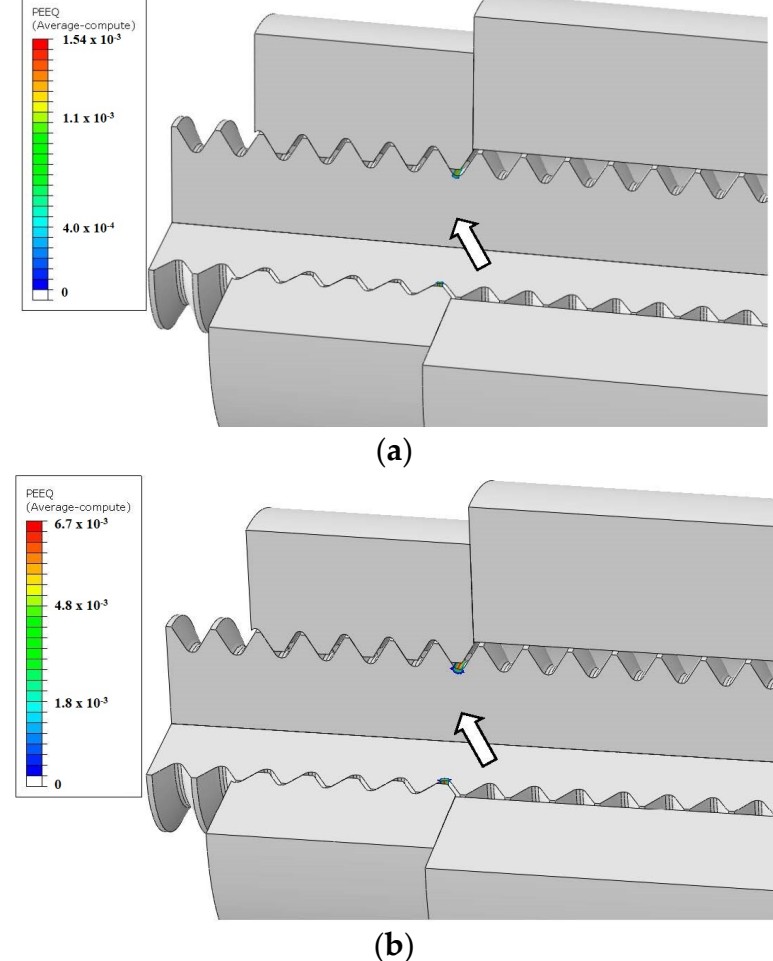

(**a**)

(**b**)

**Figure 12.** *Cont.*

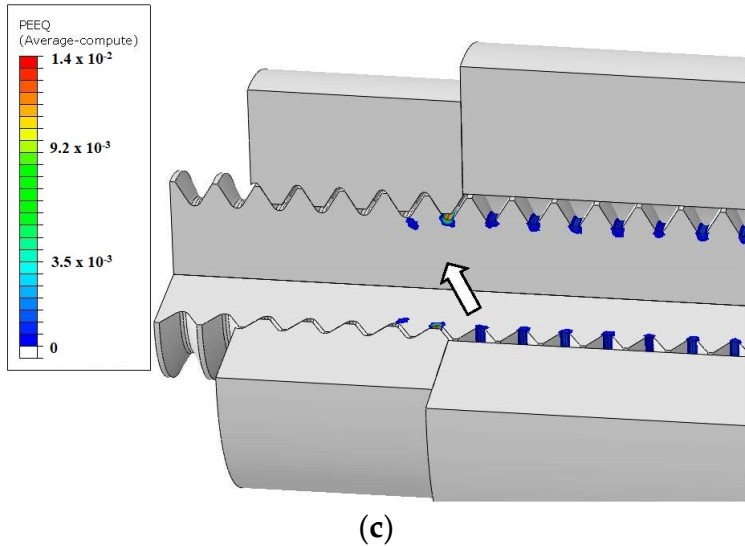

(**c**)

**Figure 12.** Strain (mm/mm) for Maximum Nominal stress of (**a**) 440.6 MPa (50% of the yield strength), (**b**) 616.8 MPa (70% of the yield strength), and (**c**) 793 MPa (90% of the yield strength).

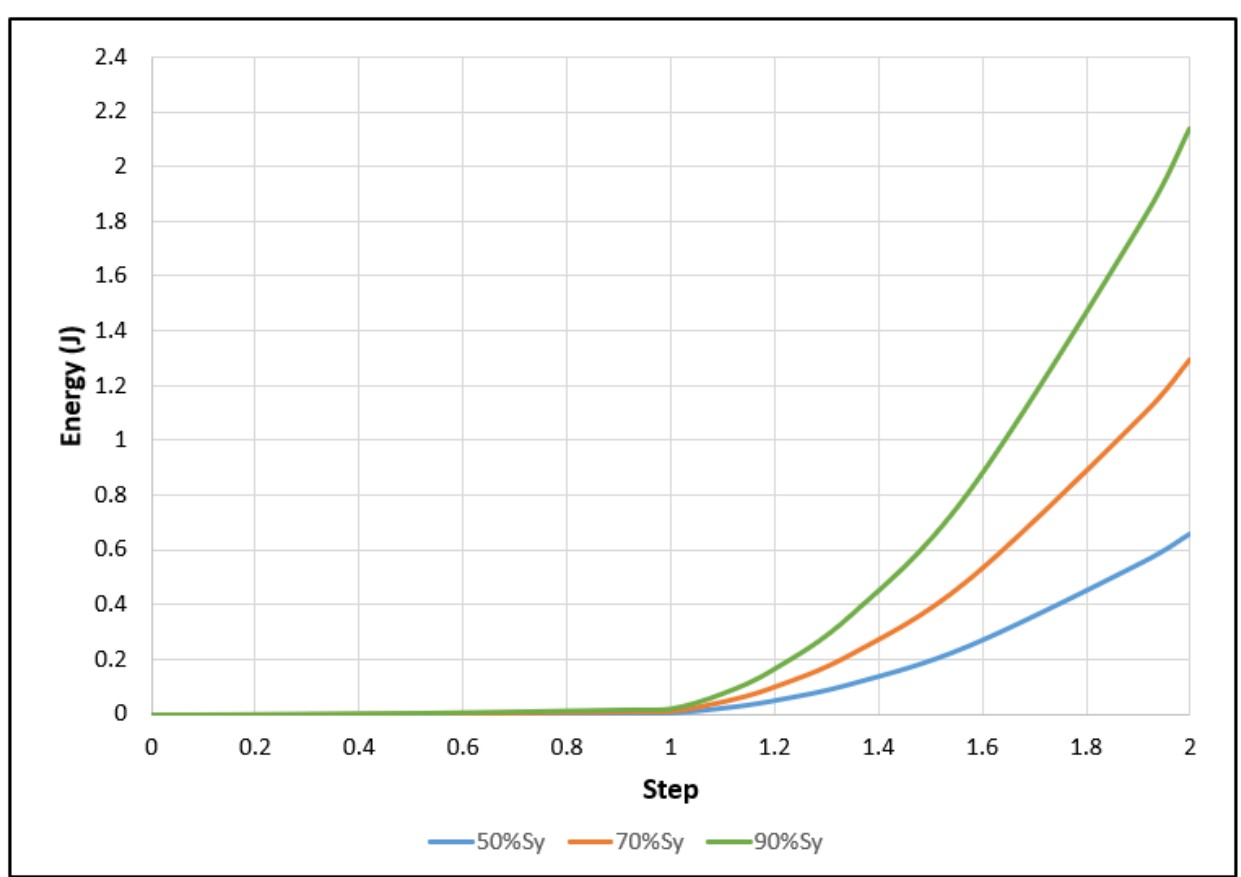

**Figure 13.** Absorbed energy (J) for a maximum nominal stress of 440.6 MPa (50% of the yield strength), 616.8 MPa (70% of the yield strength), and 793 MPa (90% of the yield strength).

For maximum nominal stress values of 440.6 MPa and 616.8 MPa, the plastic deformation was observed (Figure 12a,b) only at the last engaged thread. However, for the stress level of 793 MPa, the plastic deformation occurred in several threads (Figure 12c). This result explains the much higher strain energy absorbed for the maximum stress level and reflects the stress relaxation because more regions were under plastic deformation and,

based on the Stress-Strain curve, much more energy is absorbed in the plastic zone. The curves in Figure 13 show the absorbed energy for one cycle, where step 1 corresponds to the minimum load and step 2 to the maximum load.

Figure 14 shows the von Mises stress (MPa) for elastoplastic analysis for threaded specimens without nut interface at the situations where the maximum nominal stress values at the specimens were 440.6 MPa, 616.8 MPa, and 793 MPa, respectively. Although there is no thread engaged, a similar stress distribution was obtained for threaded specimens without nut/stud interface compared to studs. However, the stress values were reduced because there was no effect from the contact between male and female threads. Furthermore, there was no "last engaged thread," and, therefore, the maximum stress occurs at the end because of the stiffness transition, which is also explained by Saint-Venant's principle.

Figure 15 shows the strain (mm/mm) for threaded specimens without nut interface for the situations where the maximum nominal stress values of the specimens were 440.6 MPa, 616.8 MPa, and 793 MPa.

Similarly, to the conclusion observed for stud specimens, at stress levels of 440.6 MPa and 616.8 MPa, plastic deformation was observed only in the first thread, while significant plastic deformation was observed for the high-stress level of 793 MPa (90% of the yield strength).

Figure 15a,b shows in colors the regions under plastic deformation occurring at the root of the threads, where there is stiffness transition. Therefore, the probability of nucleating cracks in this region is higher. The condition with nominal stress of 90% of the yield strength (Figure 15c) resulted in more plastic deformation, which explains the stress relaxation. It is noteworthy that the ratio between the maximum and nominal stress values decreased as the maximum stress increased, which is a characteristic of self-limiting secondary stress.

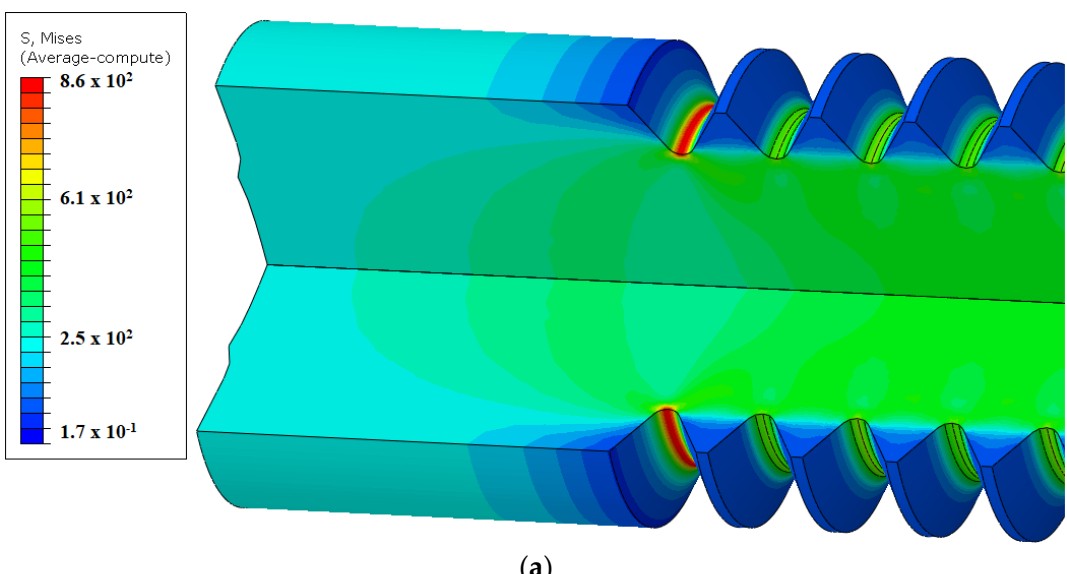

(**a**)

**Figure 14.** *Cont.*

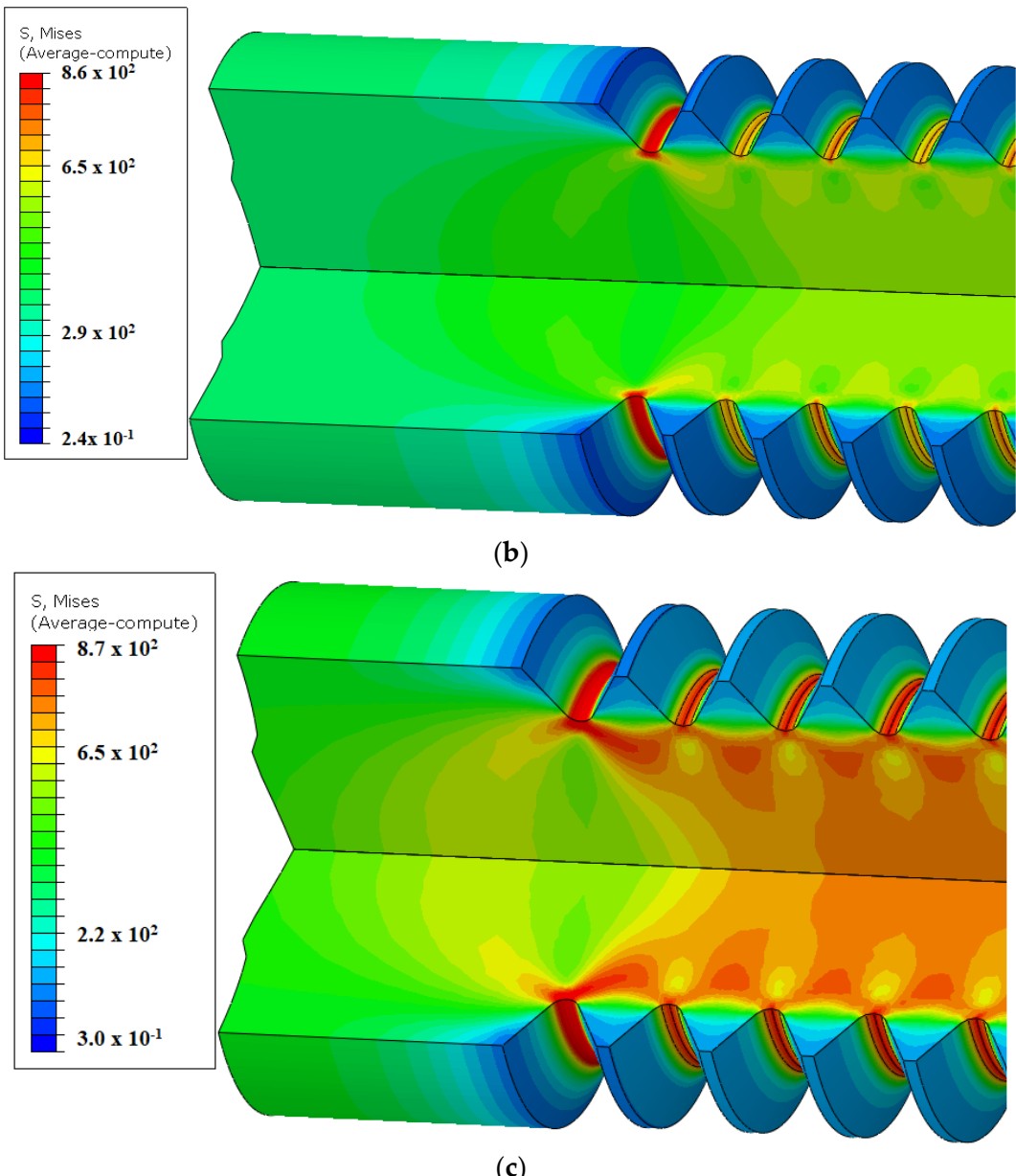

**Figure 14.** FEA for threaded specimens without nut interface. von Mises stress (MPa) for maximum nominal stress (**a**) 440.6 MPa (50% of the yield strength), (**b**) 616.8 MPa (70% of the yield strength), and (**c**) 793 MPa (90% of the yield strength).

Figures 16 and 17 show the maximum von Mises stress and maximum strain variation for both threaded and stud specimens, due to the maximum nominal stress variation. Both von Mises stress and strain increased for stud specimens as the maximum nominal stress increased. The comparison between elastoplastic finite element analysis for studs (with stud/nut interface) and threaded specimens (without nut interface) shows higher stress and strain for specimens with nut interface. The results explain the fatigue life reduction shown in Figure 9 for the studs. Figure 16 shows a comparison between threaded and stud specimens, where it was possible to realize that the von Mises stress is higher at the studs than in threaded specimens, showing the influence of the nut interface. Figure 17 shows a comparison between threaded and stud specimens where it was possible to realize that the strain is also higher at the studs than in threaded specimens.

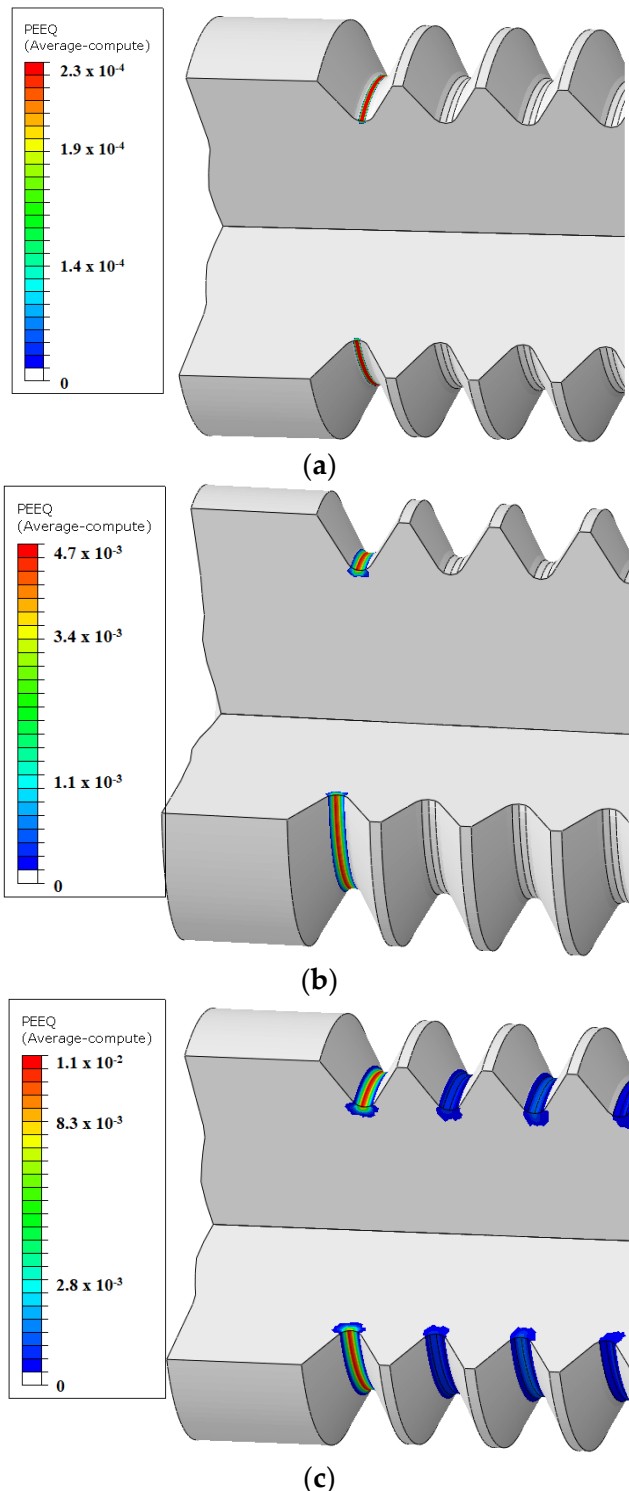

**Figure 15.** Strain (mm/mm) for threaded specimens without nut interface at a maximum nominal stress of (**a**) 440.6 MPa (50% of the yield strength), (**b**) 616.8 MPa (70% of the yield strength), and (**c**) 793 MPa (90% of the yield strength).

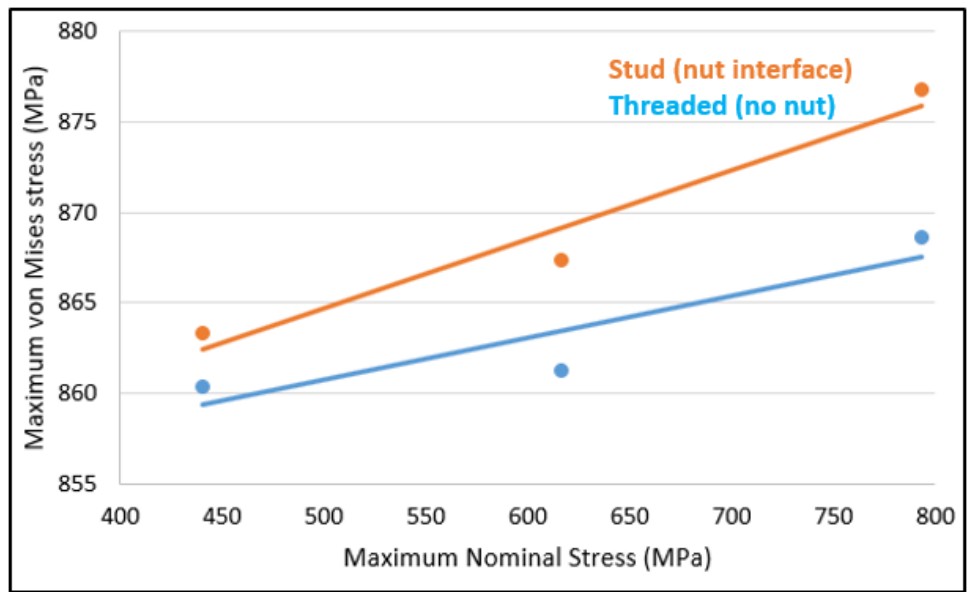

**Figure 16.** Von Mises stress (MPa) versus maximum nominal stress.

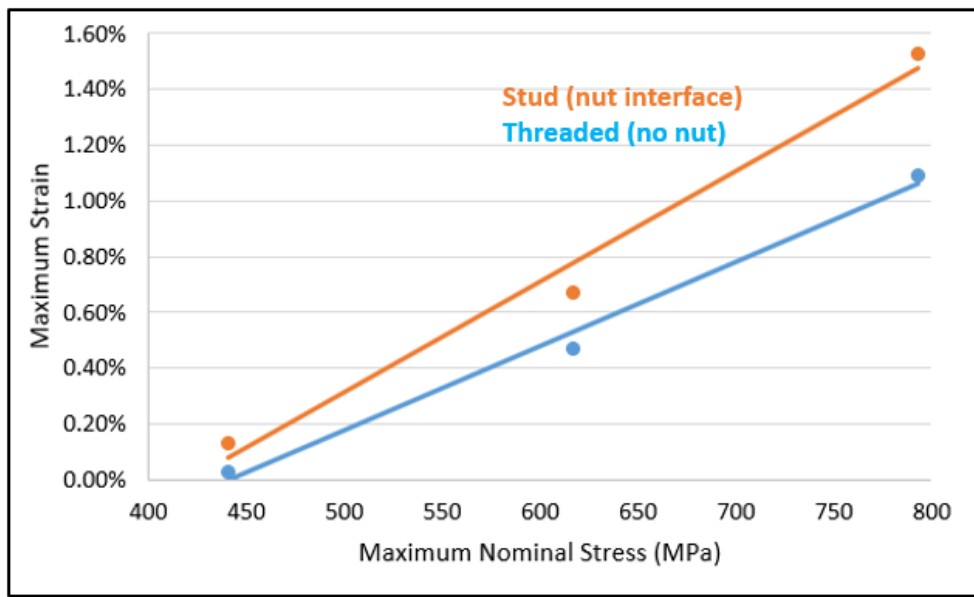

**Figure 17.** Maximum strain (mm/mm) versus maximum nominal stress.

### 3.3.2. Elastic Analysis

Elastic finite element analysis was performed to quantify the stress distribution from the thread root without nut interface. Figure 18 shows the tensile stress profile (MPa) for the situations where the maximum nominal stress was 400 MPa, 500 MPa, 600 MPa, and 700 MPa and the minimum nominal stress was 10% of the maximum. The stress concentration effect can be observed for all stress levels tested, depicting the maximum stresses near the root of the critical thread. Furthermore, the maximum distance from the root of the threads where the stress profile exceeds the nominal stress value was about 0.28 mm for all conditions.

Elastic finite element analysis was performed to quantify the stress distribution of the thread root with the nut interface. Figure 19 presents the tensile stress profile (MPa) for the situations where the maximum nominal stress was 400 MPa, 500 MPa, 600 MPa, and 700 MPa, respectively, and the minimum nominal stress was 10% of the maximum. Figure 18 shows the stress distribution for threaded specimens and Figure 19

for stud specimens, where is possible to identify that the maximum stresses values were around 2.3 times the nominal one, and the reduction to the nominal value occurs only for a distance 0.28 mm from the surface. The results show how much the thread geometry affects the stress level.

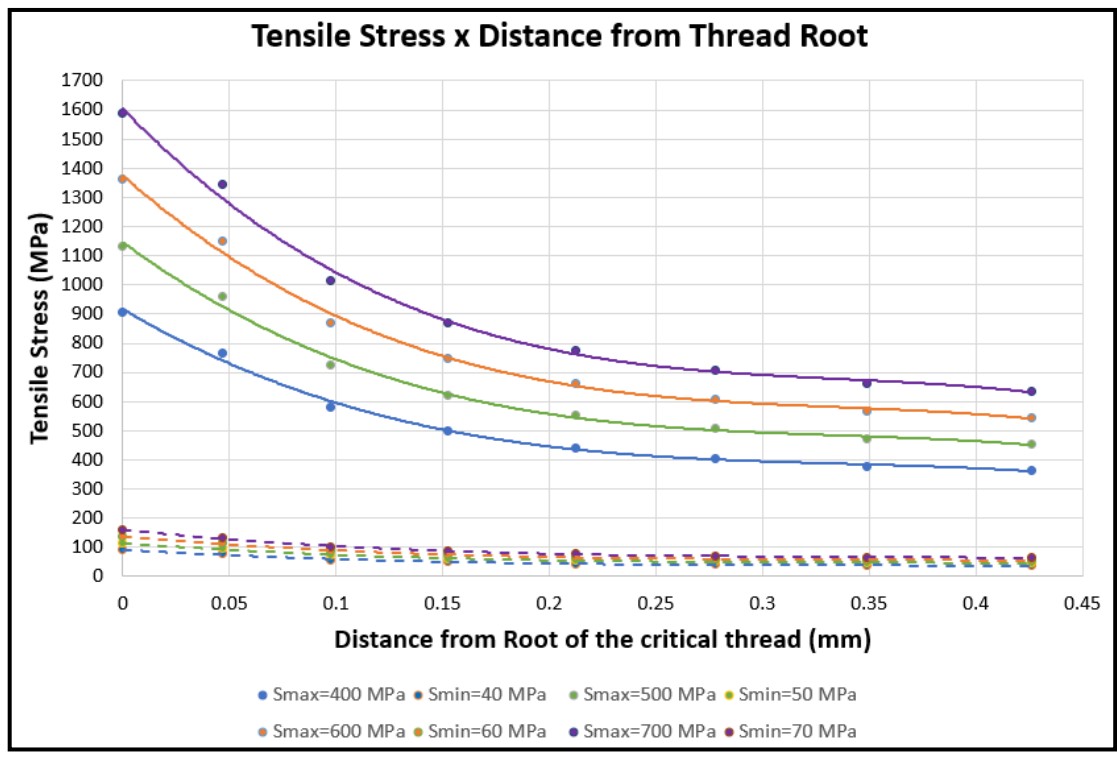

**Figure 18.** Tensile stress (MPa) from the thread root for thread specimens without the nut interface.

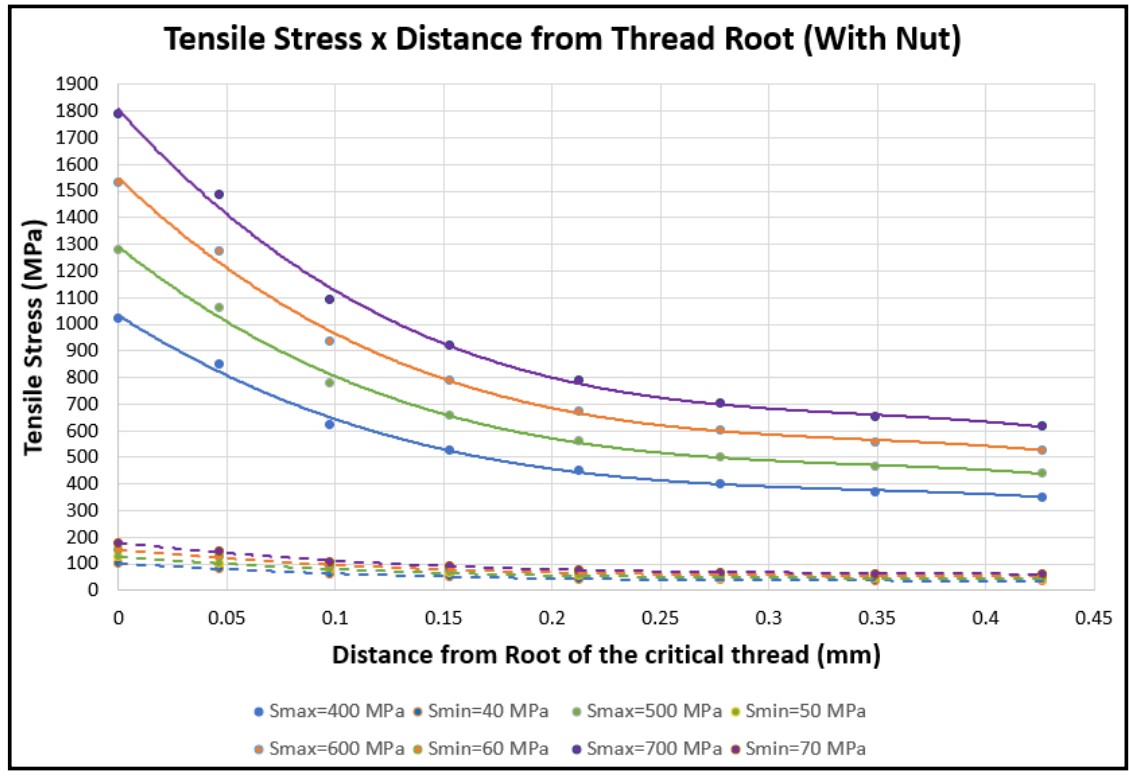

**Figure 19.** Tensile stress (MPa) from the thread root for studs.

### 3.4. Fractography

Figure 20 shows the fracture surface of standard specimens coated with Cd. The fractography shows that the cracks nucleated at the interface between the base material and the Cd coating. Figure 20 shows the nucleation site as an example of low-stress levels, where usually only one crack nucleation site was observed. At high-stress levels, as in Figure 20b, there were multiple crack nucleation sites. The crack propagation region was larger for low-stress levels since there was usually only one crack nucleation site. For all Cd coated specimens analyzed, the nucleation of cracks occurred at the interface between the Cd coating and base material. The fracture occurred with transgranular brittle crack propagation, and there was no evidence of striations or ductile features around all the specimens. The brittle crack propagation morphology is associated with the high-yield stress value and low ductility of the material studied.

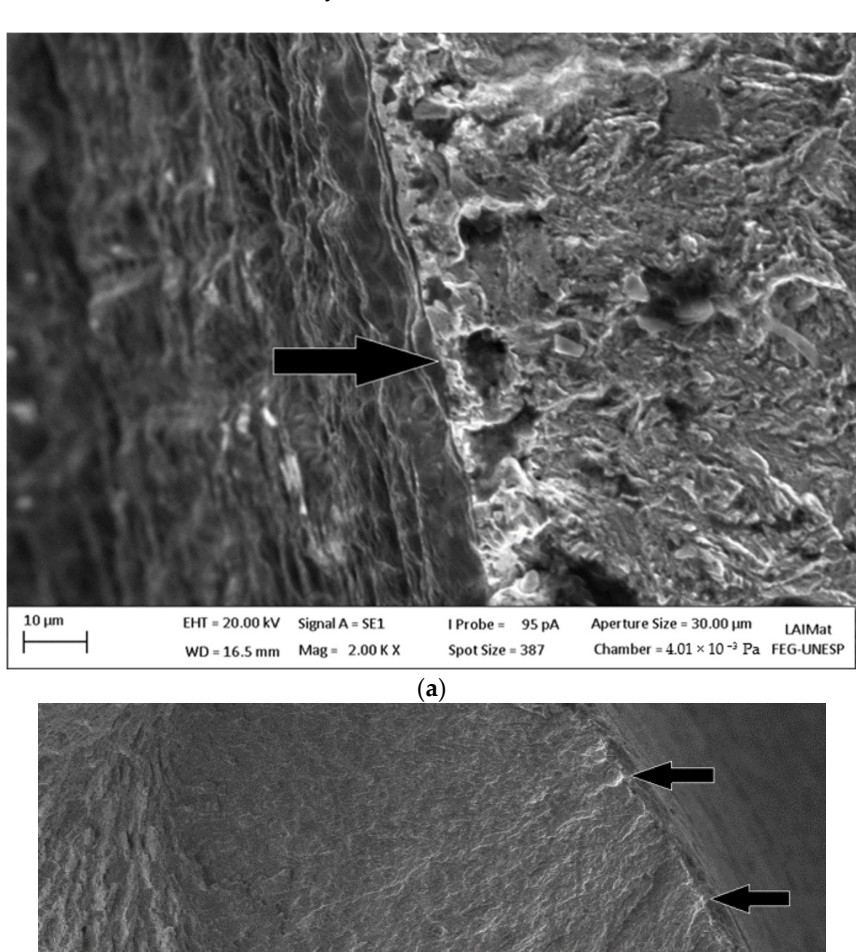

(**a**)

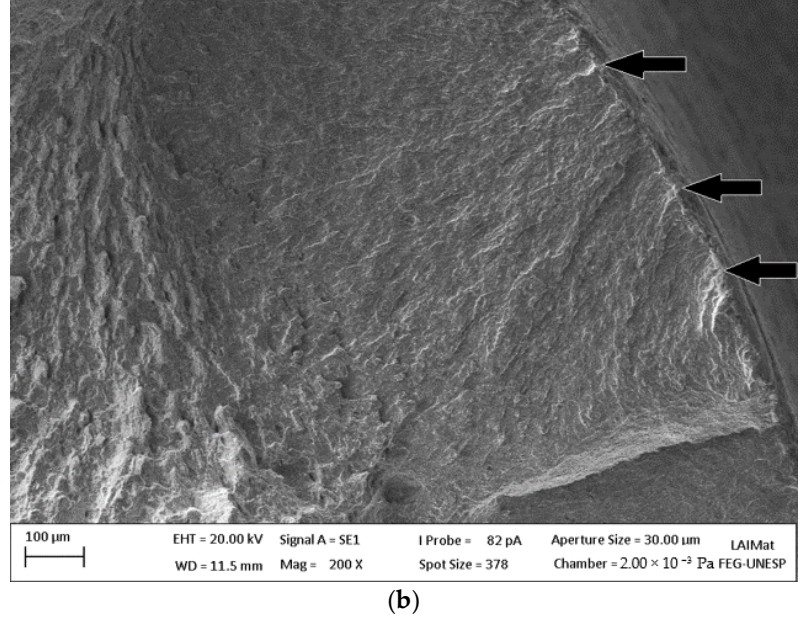

(**b**)

**Figure 20.** Fractography of Cd-coated standard specimen tested at a maximum stress of (**a**) 749 MPa $(2.06 \times 10^5$ cycles) and (**b**) 837 MPa $(3.2 \times 10^4$ cycles).

Figure 21 shows the fracture surface of a standard specimen coated with Zn-Ni. Similar to the Cd coating condition, the cracks nucleated at the interface between base metal and coating. The crack propagation features also corresponded to transgranular brittle crack propagation.

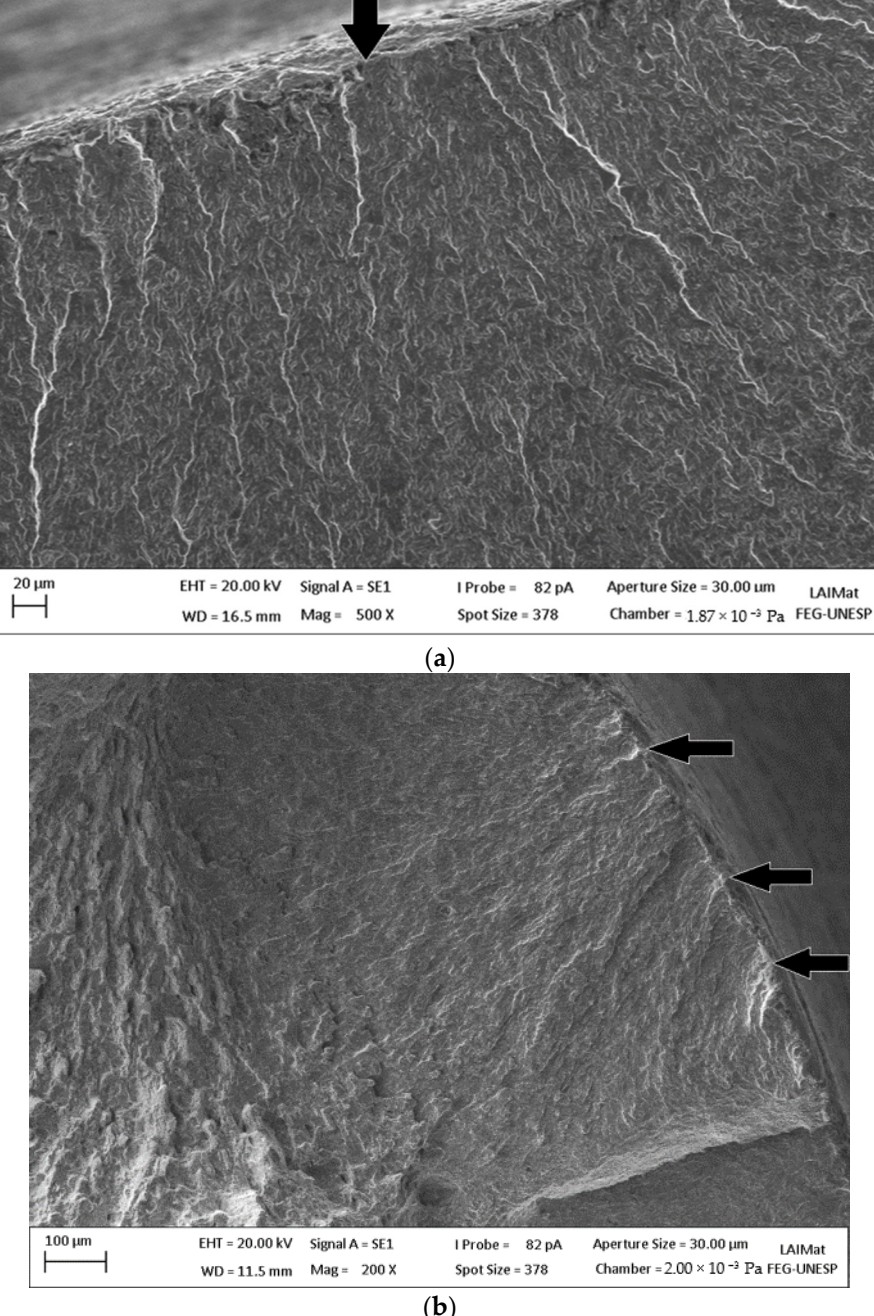

**Figure 21.** Fractography of a Zn-Ni coated standard specimen tested at a maximum stress of (**a**) 837 MPa ($2.98 \times 10^4$ cycles) and (**b**) 916 MPa ($2.08 \times 10^4$ cycles).

Figure 22 displays the typical fracture surface of threaded specimens, where it was possible to identify the nucleation of multiple cracks at the surface due to the stress concentration at the thread root. Several crack propagation fronts were observed propagating to the center direction. The fractography shows transgranular brittle crack propagation features. The final fracture occurred at the center of the specimen, where the residual

strength could not withstand the tensile loads, resulting in a fracture surface with dimples, characteristic of a ductile fracture at the center of the specimens. For all threaded specimens, the fracture surfaces were nearly similar for both coated and uncoated conditions. Since the same typical fracture surface was observed for coated specimens, the effect of stress concentration due to thread geometry was more significant than the residual stresses generated in the electrodeposition process.

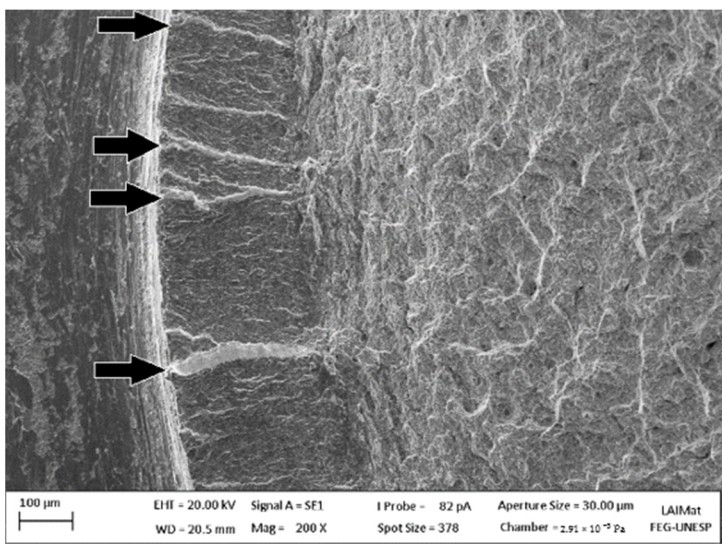

**Figure 22.** Fractography of a threaded specimen tested at a maximum stress of 793 MPa ($9.12 \times 10^3$ cycles).

## 4. Conclusions

The present work investigated numerically and experimentally the fatigue behavior of AISI 4140 steel threaded components, and the following conclusions can be drawn from the work:

- The substitution of electroplated Cd for a Zn-Ni coating is feasible regarding the axial fatigue strength. The fatigue strength of Cd and Zn-Ni threaded components was similar to the uncoated threaded base material. Therefore, there was no fatigue life debit due to the application of Cd and Zn-Ni coatings on steel threaded components. Furthermore, both smooth specimens coated with Cd and Zn-Ni have shown better fatigue life than the uncoated condition.
- The influence of the stress concentration factor was much greater than the influence of the electrodeposition process on the fatigue performance of AISI 4140 steel threaded components.
- The experimental fatigue life data showed a high contribution to reducing the fatigue life when there was contact between male and female threads, achieving a reduction of 58% with the nut/stud interface compared to threaded components without nuts.
- The studs (with stud/nut interface) had higher stress/strain values than the threaded specimens (without nut interface) in the elastoplastic FEA. The elastoplastic FEA showed the highest stress and strain concentration regions that correspond to the fatigue crack nucleation sites observed through SEM.

**Author Contributions:** Conceptualization, J.R.M.d.S. and M.F.F.; methodology, M.F.F.; software, J.R.M.d.S.; validation, V.M.d.O.V. and H.J.C.V.; formal analysis, J.R.M.d.S.; investigation, J.R.M.d.S. and M.F.F.; resources, V.M.d.O.V. and H.J.C.V.; data curation, J.R.M.d.S. and M.F.F.; writing—original draft preparation, J.R.M.d.S. and M.F.F.; writing—review and editing, J.R.M.d.S. and M.F.F.; visualization, V.M.d.O.V. and H.J.C.V.; supervision, V.M.d.O.V. and H.J.C.V.; project administration, V.M.d.O.V. and H.J.C.V.; funding acquisition, H.J.C.V. All authors have read and agreed to the published version of the manuscript.

**Funding:** FAPESP (Grant Numbers 2017/05619-0 and 2019/02125-1).

**Institutional Review Board Statement:** Not applicable.

**Informed Consent Statement:** Not applicable.

**Data Availability Statement:** Not applicable.

**Acknowledgments:** The authors gratefully acknowledge all support from São Paulo State University (UNESP) and the donated specimens by Prec-Tech.

**Conflicts of Interest:** The authors declare no conflict of interest.

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
