# Peer review of "Fatigue Analysis of Threaded Components with Cd and Zn-Ni Anticorrosive Coatings"

_metals, doi:10.3390/met11091455_

Round 1

Reviewer 1 Report

The effects of stress distribution and coating on the fatigue life of  AISI 4140 steel threaded components were studied. The influence of coating on fatigue life was obtained by fatigue test. The stress and strain distributions of threaded components under different load conditions were studied by FEA method.

  1. There are too many figures in this paper, so it is suggested that some pictures can be merged, such as Fig. 17-20, 21-23, etc.
  2. The image resolution is generally not enough, it is recommended to enhance the image resolution.
  3. Figure 10 is missing a color identifier.
  4. In Figures 13 and 14, color identifiers are suggested to be placed outside figures.
  5. There are too many items in Conclusion, it is suggested to summarize and simplify.

Author Response

  First of all, I'd like to thank you very much for all comments. It was quite important to improve the paper. If you have any further suggestion, please, just let us know.

   Bellow are shown the answer for your comments and attached is the updated paper.

  1. There are too many figures in this paper, so it is suggested that some pictures can be merged, such as Fig. 17-20, 21-23, etc.

Thank you for the remark. The Figures were merged as suggested.

  1. The image resolution is generally not enough, it is recommended to enhance the image resolution.

Thank you for the observation. The image resolution was improved.

  1. Figure 10 is missing a color identifier.

Thank you for the remark. The missing color identifier was added to the figure.

  1. In Figures 13 and 14, color identifiers are suggested to be placed outside figures.

Thank you for the observation. The color identifiers were placed outside of figures as suggested.

  1. There are too many items in Conclusion, it is suggested to summarize and simplify.

Thank you for your comment. The items in conclusion were summarized as suggested.

Best Regards,

Jefferson

Reviewer 2 Report

Re.:                 Metals-1215089

Title:                Numerical and experimental fatigue analysis of threaded components with Cd and Zn-Ni anticorrosive coatings

Authors:          Jefferson Rodrigo Marcelino dos Santos, Martin Ferreira Fernandes, Verônica Mara de Oliveira Velloso and Herman Jacobus Cornelis Voorwald

General Statement

The paper presents supplementary results for the research reported in the Ref. [17]. With no doubts some additional interesting results of experimental investigations are provided. The same concerns numerical analyses. In view of novelty of just the experimental results the research is worth to be reported in publication. However, in my opinion, the paper at its present form duplicates the methodological errors of the previous publication: there is no thematic coherence between the numerical modelling and the coating investigation and an improper nomenclature is applied to the analysed numerical problem description. The last one results in misleading title information.

In my opinion the manuscript needs major revisions. In spite of that the changes can be limited just to removing the numerical manuscript part or the changes may consist of introducing an additional clarification regarding the numerical stress and strain analysis for identification of critical joint areas the article needs modifications.

Detailed Comments/Recommendations (please refer to the attached pdf file with marked fragments)

  1. The title is misleading as the performed numerical analysis concerns strain and stress distribution modelling. The authors properly named the problem within the Abstract section. Fatigue analyses need different numerical approaches and tools. My suggestion is to modify the title and to provide within the paper an explanation that the modelling is just for indication of critical points of a crack generation.
  2. However, regarding the fact that the coating structure has not been considered within the frame of numerical model, the report can be limited just to presentation of the experimental results.
  3. Keywords section: Oil and gas were not treated directly within the manuscript.
  4. Figure 3: The presented distribution should be indicated in the figure caption.
  5. Page 6, Line 3: Indicate which specimens results are presented (bare, coated, which coating?).
  6. Page 10, Line 7: As it is difficult to spot the place in Figs 11 a and b indicate them with arrows, please.
  7. 12: Descriptions are illegible, drawings should be redrawn.

Author Response

  First of all, I'd like to thank you very much for all comments. It was quite important to improve the paper. If you have any further suggestion, please, just let us know.

   Bellow are shown the answer for your comments and attached is the updated paper.

  1. The title is misleading as the performed numerical analysis concerns strain and stress distribution modelling. The authors properly named the problem within the Abstract section. Fatigue analyses need different numerical approaches and tools. My suggestion is to modify the title and to provide within the paper an explanation that the modelling is just for indication of critical points of a crack generation.

We are very grateful for your comments. The title was updated. We hope that the new title is capable of properly meet the expectations of the readers. Furthermore, an explanation that the modelling aims mainly to indicate the critical crack nucleation points was included in the paper as suggested.

2. However, regarding the fact that the coating structure has not been considered within the frame of numerical model, the report can be limited just to presentation of the experimental results.

Thank you for your insightful comments. We understand that the coating structure was only analyzed through the experimental analysis and not through the modelling. This paper aims to study the application of the steel studs in petrochemical companies, where both the coating and the stress concentration effect play a significant role in the fatigue performance. Therefore, the finite element analysis aims to clarify the effect of stress concentration in those components since studs and threaded specimens were tested. We included an explanation in the paper to clearly state the aims of experimental and modelling analysis.

3. Keywords section: Oil and gas were not treated directly within the manuscript.

Thank you for the remark. The keyword section was updated.

4. Figure 3: The presented distribution should be indicated in the figure caption.

Thank you for the remark. The figure caption was updated.

5. Page 6, Line 3: Indicate which specimens results are presented (bare, coated, which coating?).

Thank you for the observation. The text was modified as suggested.

6. Page 10, Line 7: As it is difficult to spot the place in Figs 11 a and b indicate them with arrows, please.

Thank you for your comment. The figure was updated with arrows to clearly indicate the spots.

7. Descriptions are illegible, drawings should be redrawn.

Thank you for the remark. The Figures were updated.

Best Regards,

Jefferson

Reviewer 3 Report

The paper is well written but it needs some improvements concerning both the paper organization and the discussion:

  • paragraph 2.2: please, add details about the loading waveform;
  • paragraph 3.1: it is necessary to modify the organization of the paper. Firstly, it is necessary to describe the investigated materials and, then, the fatigue results.
  • subparagraph 3.1.1: considering the results statistical dispersion, some comments are necessary;
  • subparagraph 3.1.3, Figs. 7: Larger magnification micrographies are necessary. The figures quality is too low; scale magnification bars must be more evident;
  • subparagraph 3.1.3, Figs. 8: etching quality is too low. Phases in coatings are not evident and clearly described.  Scale magnification bars must be more evident;
  • Paragraph 3.2: Numerical results validation is missing;
  • Paragraph 3.4, Figs 25-27: Arrows in the fractographies are necessary in order to connect the description in the text with the figures.
  • A discussion of the damaging mechanisms is missing. In the present form, the paper is merely phenomenological.

Author Response

   First of all, I would like to thank you so much for all comments that was quite important to improve the paper. Please, have a look below for our answer and the updated paper attached. Feel free to ask for another improvement. 

Thanks and Best Regards,

Jefferson

  • paragraph 2.2: please, add details about the loading waveform;

Thank you for your observations. The waveforms were sinusoidal with frequency of 20 Hz. The details about the loading waveforms were added.

  • paragraph 3.1: it is necessary to modify the organization of the paper. Firstly, it is necessary to describe the investigated materials and, then, the fatigue results.

Thank you for your comments. The organization of the paper was modified to properly describe the investigated materials before fatigue data.

  • subparagraph 3.1.1: considering the results statistical dispersion, some comments are necessary;

Thank you for the remark. Additional comments regarding the statistical dispersion of the fatigue data were provided.

  • subparagraph 3.1.3, Figs. 7: Larger magnification micrographies are necessary. The figures quality is too low; scale magnification bars must be more evident;

Thank you for the remark. A microstructure with larger magnification was added to item “a” of the Figure. The scale bars were modified.

  • subparagraph 3.1.3, Figs. 8: etching quality is too low. Phases in coatings are not evident and clearly described.  Scale magnification bars must be more evident;

Thank you for the remark. Further indications in the figures were added and the scale bars were modified.

  • Paragraph 3.2: Numerical results validation is missing;

Thank you for the important comment. An explanation that correlates the numerical and experimental results was added to show the accuracy of the finite element analysis results.

  • Paragraph 3.4, Figs 25-27: Arrows in the fractographies are necessary in order to connect the description in the text with the figures.

Thank you for the observation. The arrows were added to the fractographies.

  • A discussion of the damaging mechanisms is missing. In the present form, the paper is merely phenomenological.

Thank you for the insightful comment. Further explanations were provided throughout the manuscript to try to improve the damage mechanisms discussions.

Reviewer 4 Report

It is an interesting paper but some additionnal  work is needed for the presentation, for the as received material description , for the description of the fatigue tests, machine and gripping system, for the introduction of the forces in the model, description and analysis of the crack nucleation and propagation. A final discussion to show the benefit of the finite elements simulation is missing...

1.Introduction. First sentence on page 1 should be improved : a large number of studs to withstand internal pressures of 22,500 psi. Make clear that the  components (pipes, vessels etc..) are loaded  to very high pressures not the studs. The studs are loaded with stresses comprising  safety margins. 

page 1, line 5, please write in words the meaning of acronym FEA

You could improve the last paragraph of the introduction to be more specific on the fatigue tests (type of test, temperature etc..) conducted and then add a justification why you did the FEA analysis

2. Materials and methods.

2.1 Materials and coatings.

Please indicate the metallurgical condition of the material before machining the specimens and in which form it was delivered ( plate, rod?).A description of the as received microstructure is missing. This must be added.)

Your description of the heat treatment is not correct . You did an homogenization at 900°C for 3 h followed by a quenching in oil at 81°C and then a double tempering treatment. Your mechanical parameters are very high (860 MPa and 978 MPa , yield  and ultimate stress respectively). 

If the heat treatments were done by you on the machined specimens, how did you determine the tensile parameters ? Did you conduct  tensile tests on the smooth specimens? Give a reference or explanation....

End of page 2. You should invert the sentences and explain  first that you used optical microscopy and then that you etched the specimens to reveal the microstructure

You should also indicate here that you did SEM analysis and indicate instrument and type of observation

2.2 Experimental fatigue tests. Page 3. You write conditions better would specimen configurations. You tested using the same mechanical  condition, load controlled tests, room temperature, 20Hz and  different surface coatings.

Please indicate testing machine type and gripping system used. Your specimen heads are cylindrical and button head, how did you grip the specimens? What was the torque applied to the nut/stud installation?

Scanning electron microscopy  sentence belongs to preceding chapter

3.1.1 Fatigue data.

First indicate how you derived the stress for thread and smooth geometry.

Figure 4. Please use larger symbols and thicker trend lines. One can see a huge dispersion of the data points. Why such a large difference between the two coatings conditions?

Figure 5 Again larger symbols and lines, legend larger, will improve the quality of the plot

Figure 6. The installation of  a nut degrades the fatigue life. So what is the influence of the  torque applied to the nut, did you test different installation torques? Describe the location of the crack inducing failure for both geometries! Did you observe multiplle cracks?

3.1.3. Microstrucure and coating characterization This part belongs before the fatigue tests part. First present the microstructure then the fatigue results.

The micrographs  Fig 8  should indicate more clearly the zone corresponding to the coating, using arrows or marks. Specially fig 8 b is not good enough, please replace it with a better one.

3.2.1 Elastoplastic analysis 

You should clearly indicate how the forces are introduced in your assembly. This is the  same lack of description as noted for the gripping system used in the tests (par 2.2). Without this explanation, it is difficult to understand your results. Do you consider some pre-load between the nuts shown in Fig.11?

On page 10, first paragraph you claim the existence of high plastic strains but the scale in your Fig. 11 a,b indicates strains of the  order of the elastic limit, also still elastic strains... This is a contradiction . Plasticity seems to occur to a large extent  only for case c

Combine Fig 12, a,b and c into a single figure, to save space.

Regarding the calculation of the absorbed energy , are you refering to the energy absorbed by one cycle? What is the meaning of the step parameter?

3.3.2 Elastic analysis.

Combine Figs 17,18,19 and 20 into a single figure, to save space.

Combine Figs 21,22,23 and 24 into a single figure, to save space

Your statement on page 18, bottom of paragraph:  The results show the importance of the thread geometry...  Seems strange since you use for all calculations and tests the  same thread UNC 1/4-20

The stress increase at the root is not only a 'geometry' effect but reflects also the decrease of the bearing section at the thread root

The elastic distribution confirms an obvious fact, the crack is expected to nucleate at the thread root. 

What for did you makee the FEA analysis?  Was it helpful to understand the fatigue tests? What did you learn from the calculations, which was unexpected? A little discussion about that is missing

3.4 Fractography

You do not describe how the failures occured  in your test. You should present a diagram showing the last cycles of a test for the three situations, smooth , thread with and without nut. How many cracks did you observed on the surface of the failed specimens, for the different situations. At which location did they start? Your fractographs seems to indicate a transgranular brittle crack propagation. No evidence of striations or other ductile features. How was the crack propagating ? Did you observe a continuous brittle crack around all the circumference? Your fractographs are for low number of cycles failures (max 32828 cycles) but in your Fig. 4,5 and 6, some specimens last over 105 cycles, was the brittle part of the crack larger in this case?

4. Conclusions

Your third conclusion: The fatigue strength ....was similar to the base material....?  Would be better to be more precise : than uncoated threaded material. ( Fig. 4 shows an effect of improvement for smooth surface)

Conclusion number 4: you did not discuss the effect of the pre-load of the assembly in the paper. 

Conclusion number 5: it is not demonstrated in the paper! The conclusion is probably obvious for you but you need to show some piece of evidence . See remarks on the fractography.

Author Response

 First of all, thanks very much for all your comments that was quite important to improve our paper. Below is shown the answer for each comment and attached is the updated paper. Feel free to ask for any further improvement. 

Best Regards,

Jefferson

1.Introduction. First sentence on page 1 should be improved : a large number of studs to withstand internal pressures of 22,500 psi. Make clear that the  components (pipes, vessels etc..) are loaded  to very high pressures not the studs. The studs are loaded with stresses comprising  safety margins. 

Thank you for the remark. The sentence was modified.

page 1, line 5, please write in words the meaning of acronym FEA

Thank you for the remark. The Finite Element Analysis (FEA) term was added.

You could improve the last paragraph of the introduction to be more specific on the fatigue tests (type of test, temperature etc..) conducted and then add a justification why you did the FEA analysis.

Thank you for the remark. The last paragraph of the introduction was modified.

  1. Materials and methods.

2.1 Materials and coatings.

Please indicate the metallurgical condition of the material before machining the specimens and in which form it was delivered ( plate, rod?).A description of the as received microstructure is missing. This must be added.)

Thank you for the remark. The metallurgical and microstructural condition of the material was provided.

Your description of the heat treatment is not correct . You did an homogenization at 900°C for 3 h followed by a quenching in oil at 81°C and then a double tempering treatment. Your mechanical parameters are very high (860 MPa and 978 MPa , yield  and ultimate stress respectively). 

Thank you very much for your comments. The correct description of the heat treatment was provided as you suggested.

If the heat treatments were done by you on the machined specimens, how did you determine the tensile parameters ? Did you conduct  tensile tests on the smooth specimens? Give a reference or explanation....

We performed tensile tests on smooth specimens. The explanation was added to the manuscript.

End of page 2. You should invert the sentences and explain  first that you used optical microscopy and then that you etched the specimens to reveal the microstructure.

Thank you for the remark. The text was modified.

You should also indicate here that you did SEM analysis and indicate instrument and type of observation

Thank you for the remark. The SEM analysis was indicated and the model of the instrument and type of observation were provided.

2.2 Experimental fatigue tests. Page 3. You write conditions better would specimen configurations. You tested using the same mechanical  condition, load controlled tests, room temperature, 20Hz and  different surface coatings.

Thank you for the remark. The term “conditions” was replaced by “specimen configurations”.

Please indicate testing machine type and gripping system used. Your specimen heads are cylindrical and button head, how did you grip the specimens? What was the torque applied to the nut/stud installation?

Thank you for the remark. Further information was provided regarding machine type and gripping system used.

Scanning electron microscopy  sentence belongs to preceding chapter

Thank you for the remark. The SEM sentence was moved to the preceding chapter.

3.1.1 Fatigue data.

First indicate how you derived the stress for thread and smooth geometry.

Thank you for your comment. We have added a note specifying the stress was derived from the minimum diameter to compute the real dimension for each specimen.

Figure 4. Please use larger symbols and thicker trend lines. One can see a huge dispersion of the data points. Why such a large difference between the two coatings conditions?

Thanks for this important question. We have improved the chart and added information regarding the fatigue life difference between the two coating conditions that could be due to different compression stress and coating thickness.

Figure 5 Again larger symbols and lines, legend larger, will improve the quality of the plot

Thanks very much for your comment. We have improved the quality.

Figure 6. The installation of  a nut degrades the fatigue life. So what is the influence of the  torque applied to the nut, did you test different installation torques? Describe the location of the crack inducing failure for both geometries! Did you observe multiplle cracks?

Thanks for your comment. Regarding the installation torque, we have added information specifying that no torque was applied. Regarding the location of cracks, we observed that the cracks started at the coating interface with base material for smooth specimen and from the surface with multiple cracks for threaded specimens. Furthermore, the location of cracks (i.e., failure) match with the FEA for threaded specimens.

3.1.3. Microstrucure and coating characterization This part belongs before the fatigue tests part. First present the microstructure then the fatigue results.

Thank you for the suggestion. The “microstructure and coatings characterization” section was moved to before the fatigue results.

The micrographs  Fig 8  should indicate more clearly the zone corresponding to the coating, using arrows or marks. Specially fig 8 b is not good enough, please replace it with a better one.

Thank you for the remark. The pictures have been updated with arrows and marks to clearly indicate the coatings.

3.2.1 Elastoplastic analysis 

You should clearly indicate how the forces are introduced in your assembly. This is the  same lack of description as noted for the gripping system used in the tests (par 2.2). Without this explanation, it is difficult to understand your results. Do you consider some pre-load between the nuts shown in Fig.11?

Thanks very much for this important comment. Regarding the testing machining, we have added information regarding the Instron 8801 equipment that uses hydraulic wedge gripping system, and the specimens attached regions was also considered in the FEA and specified in Figure 4. The pre-load was not considered in this study.

On page 10, first paragraph you claim the existence of high plastic strains but the scale in your Fig. 11 a,b indicates strains of the  order of the elastic limit, also still elastic strains... This is a contradiction . Plasticity seems to occur to a large extent  only for case c

Thank you very much. We have updated the sentence for an accurate discussion and to avoid this contradiction.

Combine Fig 12, a,b and c into a single figure, to save space.

Thanks for the excellent suggestion. We have combined as suggested.

Regarding the calculation of the absorbed energy , are you refering to the energy absorbed by one cycle? What is the meaning of the step parameter?

Thanks very much. We have added information showing the energy is for one cycle where step 1 means minimum load and step 2 means maximum load.

3.3.2 Elastic analysis.

Combine Figs 17,18,19 and 20 into a single figure, to save space.

Thanks very much. We have implemented this good suggestion.

Combine Figs 21,22,23 and 24 into a single figure, to save space

Thanks very much. We have implemented this good suggestion.

Your statement on page 18, bottom of paragraph:  The results show the importance of the thread geometry...  Seems strange since you use for all calculations and tests the  same thread UNC 1/4-20

Thanks for the good suggestion. We have improved the text to better describe what we need, regarding how much the geometry affect in a comparison of the coating process.

The stress increase at the root is not only a 'geometry' effect but reflects also the decrease of the bearing section at the thread root

The elastic distribution confirms an obvious fact, the crack is expected to nucleate at the thread root. 

What for did you makee the FEA analysis?  Was it helpful to understand the fatigue tests?

What did you learn from the calculations, which was unexpected? A little discussion about that is missing

Thanks very much for all excellent comments shown above. The reason why FEA was performed is to allow future understanding and extrapolation using only simulations because is not always feasible to perform physical tests for all thread specification but of course, care should be taken, and appropriate safety factor should be considered.

3.4 Fractography

You do not describe how the failures occured  in your test. You should present a diagram showing the last cycles of a test for the three situations, smooth , thread with and without nut. How many cracks did you observed on the surface of the failed specimens, for the different situations. At which location did they start? Your fractographs seems to indicate a transgranular brittle crack propagation. No evidence of striations or other ductile features. How was the crack propagating ? Did you observe a continuous brittle crack around all the circumference? Your fractographs are for low number of cycles failures (max 32828 cycles) but in your Fig. 4,5 and 6, some specimens last over 105 cycles, was the brittle part of the crack larger in this case?

Thank for all of your remarks regarding fractography. Additional SEM images were added for lower stress levels as suggested (about 2x105 cycles). The discussion was improved based on the comments.

  1. Conclusions

Your third conclusion: The fatigue strength ....was similar to the base material....?  Would be better to be more precise : than uncoated threaded material. ( Fig. 4 shows an effect of improvement for smooth surface)

Thanks very much for your comment. The text was changed as suggested.

Conclusion number 4: you did not discuss the effect of the pre-load of the assembly in the paper. 

Thanks for this important comment. We have improved the text to inform that no torque was applied for pre-load.

Round 2

Reviewer 2 Report

Re.:              Metals-1215089_v2

Title:             Fatigue analysis of threaded components with Cd and Zn-Ni anticorrosive coatings

Authors:        Jefferson Rodrigo Marcelino dos Santos, Martin Ferreira Fernandes, Verônica Mara de Oliveira Velloso and Herman Jacobus Cornelis Voorwald

General Statement

As the most of the indicated problems have been explained and the manuscript has been properly corrected I am recommending the paper for publication in spite of that I am still not convinced of the necessity for presentation of its numerical part at a whole. With no doubt the results are original but, in my opinion, they do not contribute much to the discussed problem. I personally would recommend to account for different properties of a thin surface layer at least or for performing a professional numerical fatigue analysis. Nevertheless I understand that opinions can differ and I am grateful that the authors have taken up the discussion.

Summarizing, my recommendation is accept at present form.

Author Response

Thank you very much for all your support and valuable comments during the review process.

Reviewer 3 Report

The paper can be published

Author Response

Thank you very much for all your support. We really appreciated your suggestions that improved the paper. Attached is the report regarding English review.

Best Regards,

Jefferson

Reviewer 4 Report

Dear  author ,

Thank you for the improvements.

Abstract: indicate that the fatigue testing was done in air at RT ( you never precise it)

Introduction:

The first sentence is too long . I propose to change as follows:

Oil and gas are extracted from nature under  high pressures. The transport from the well to the platform requires equipment like  wellhead, Christmas tree, PLET, PLEM, manifolds and pipes which need to withstand internal pressures up to 25'000 psi. Any failure could be catastrophic for  environment , people and oil company.

and further at the end, it is better to do two sentences again:

The FEA  aims mainly to identify the  critical crack nucleation points on threaded components . The coating effect was not adressed in simulations.

and further: 

a frequency of 20Hz

and further:

Finite element analysis was used to describe the stress distribution...

Materials  and methods:

The microstructure of the  material ( delete: in this work ) is tempered martensite

Your environment is sea water! you should tell that your testing was done in air! so you do not have the correct environment, but if you tell it at least we know that extrapolation to the water situation can be tricky. A little remark about that should be added !

2.2 Experimental fatigue tests.

May be you could precise more:

it is noteworthy that no torque was applied between nut and sleeve.

2.3  Finite element analysis:

second paragraph, proposal to improve the text :

The finite element analysis  (FEA) was used to investigate the stress distribution of a particular configuration of threaded component and to correlate it with the experimental results. Based on this comparison, the behavior of other geometries can be predicted and a lot of experimental work can be saved.

3.2 Fatigue data

Precise the test environment and temperature somewhere !

line 9 of first paragraph: change taking to taken

line 10 of second paragraph:

You could remove the word  coated, seems not necessary.

In Figure 8, you changed the legend and both conditions are written as "no nut" . Please check and correct if necessary, because in the text you mention : sleeve nut interface. In fact as there is no pre-loading, the nut should not have a big influence but later your calculations indicate that it is important (location with high strain at the interface).

3.3. Finite element analysis

page 12, first paragraph, it is necessary to improve the text (make two sentences):

line 3: ...yield strength. Even for a nominal stress of 50% and 70% of the yield, a small region at the bottom of the thread reached  plasticity, due to stress concentration. 

On page 16 , first paragraph:

The curves of Figure 12 show the absorbed energy for one cycle, where step 1 corresponds to the minimum  load and step to to the maximum  load.

3.3.2 Elastic analysis

On page 20, last paragraph, your conclusion that the coatings are not negatively influencing the fatigue life because the zone of high stress (higher than nominal) is only 0.28 mm is not generally correct. You should remove it.

Or you want to say that the difference due to coatings observed on smooth specimens does not show up with a threaded specimen ?

For instance, it is known that hard coatings are very bad for the fatigue life of aluminium components because they can break and initiate stress concentrations which nucleate a crack. A 'bad' coating could cause problems at low stresses.

3.4 Fractography 

The description done and the fractographs  indicate a brittle crack propagation.  On page 10 you indicate that the hydrogen embrittlement did not occur. Why? It is difficult to judge if hydrogen is responsible for your case but the micrograph 19 a  indicates a fracture mode known as quasi cleavage, facets, cleaved grains at some places, other places more ductile, no striations. This type of rupture facies is also reported for hydrogen embrittled steels. Room temperature and air testing , low stresses are prone to develop such fracture surfaces. Humidity in air is enough to generate some hydrogen.

But these effects are also due to the high stress values of the material (yield stress 820MPa!). If you have it , you could report the materials ductility ( from your tensile tests on smooth material, in section 2.1). If the ductility is low , your material will have tendency for brittleness . Regarding hydrogen, you could check following reference : P.J.Cotterill et al., Acta metall. mater. 40, 1992, page 2753-2764

If you work on the above remarks, your paper is OK for publication

Author Response

Thank you very much for all your extremely valuable comments and insights during the review process that significantly improved the paper. Furthermore, all your following remarks were addressed as suggested.

Thank you for the improvements.

Abstract: indicate that the fatigue testing was done in air at RT ( you never precise it)

Thank you for the remark. The text was updated to “Axial fatigue tests at room temperature […] were performed…”

Introduction:

The first sentence is too long . I propose to change as follows:

Oil and gas are extracted from nature under  high pressures. The transport from the well to the platform requires equipment like  wellhead, Christmas tree, PLET, PLEM, manifolds and pipes which need to withstand internal pressures up to 25'000 psi. Any failure could be catastrophic for  environment , people and oil company.

Thank you very much for the comment. The text was changed as suggested.

and further at the end, it is better to do two sentences again:

The FEA  aims mainly to identify the  critical crack nucleation points on threaded components . The coating effect was not adressed in simulations.

Thank you for the remark. The text was modified as suggested.

and further: 

a frequency of 20Hz

Thank you for the remark. The text was modified as suggested.

and further:

Finite element analysis was used to describe the stress distribution...

Thank you for the remark. The text was modified as suggested.

Materials  and methods:

The microstructure of the  material ( delete: in this work ) is tempered martensite

Thank you for the remark. The text was modified as suggested.

Your environment is sea water! you should tell that your testing was done in air! so you do not have the correct environment, but if you tell it at least we know that extrapolation to the water situation can be tricky. A little remark about that should be added !

Thank you very much for the valuable suggestion. We added a remark to inform that the tests were performed in air, and additional comments regarding air and subsea environments were provided.

2.2 Experimental fatigue tests.

May be you could precise more:

it is noteworthy that no torque was applied between nut and sleeve.

Thank you for the remark. The text was modified as suggested.

2.3  Finite element analysis:

second paragraph, proposal to improve the text :

The finite element analysis  (FEA) was used to investigate the stress distribution of a particular configuration of threaded component and to correlate it with the experimental results. Based on this comparison, the behavior of other geometries can be predicted and a lot of experimental work can be saved.

Thank you very much. The text was improved as suggested.

3.2 Fatigue data

Precise the test environment and temperature somewhere !

Thanks a lot. The terms “at room temperature and air environment” were added.

line 9 of first paragraph: change taking to taken

Thank you for the remark. The text was modified as suggested.

line 10 of second paragraph:

You could remove the word  coated, seems not necessary.

Thank you for the remark. The text was modified as suggested.

In Figure 8, you changed the legend and both conditions are written as "no nut" . Please check and correct if necessary, because in the text you mention : sleeve nut interface. In fact as there is no pre-loading, the nut should not have a big influence but later your calculations indicate that it is important (location with high strain at the interface).

Thank you very much for the comment. We have updated the legend. This graph shows the difference between the threaded specimen (without nut) and stud (with nut interface).

3.3. Finite element analysis

page 12, first paragraph, it is necessary to improve the text (make two sentences):

Thank you for the remark. The text was modified as suggested.

line 3: ...yield strength. Even for a nominal stress of 50% and 70% of the yield, a small region at the bottom of the thread reached  plasticity, due to stress concentration. 

Thank you for the remark. The text was modified as suggested.

On page 16 , first paragraph:

The curves of Figure 12 show the absorbed energy for one cycle, where step 1 corresponds to the minimum  load and step to to the maximum  load.

Thank you for the remark. The text was modified as suggested.

3.3.2 Elastic analysis

On page 20, last paragraph, your conclusion that the coatings are not negatively influencing the fatigue life because the zone of high stress (higher than nominal) is only 0.28 mm is not generally correct. You should remove it.

Thanks for the suggestion. We have updated the text.

Or you want to say that the difference due to coatings observed on smooth specimens does not show up with a threaded specimen?

Thank you for the comment. Exactly. The fatigue behavior difference due to coatings observed on smooth specimens does not show up with a threaded specimen since the thread geometry affects the stress level at the surface close to the root.

For instance, it is known that hard coatings are very bad for the fatigue life of aluminium components because they can break and initiate stress concentrations which nucleate a crack. A 'bad' coating could cause problems at low stresses.

Thank you for the explanation. We completely agree with this statement.

3.4 Fractography 

The description done and the fractographs  indicate a brittle crack propagation.  On page 10 you indicate that the hydrogen embrittlement did not occur. Why? It is difficult to judge if hydrogen is responsible for your case but the micrograph 19 a  indicates a fracture mode known as quasi cleavage, facets, cleaved grains at some places, other places more ductile, no striations. This type of rupture facies is also reported for hydrogen embrittled steels. Room temperature and air testing , low stresses are prone to develop such fracture surfaces. Humidity in air is enough to generate some hydrogen.

But these effects are also due to the high stress values of the material (yield stress 820MPa!). If you have it , you could report the materials ductility ( from your tensile tests on smooth material, in section 2.1). If the ductility is low , your material will have tendency for brittleness . Regarding hydrogen, you could check following reference : P.J.Cotterill et al., Acta metall. mater. 40, 1992, page 2753-2764

Thanks very much for all explanation. We have replaced the sentence  “Furthermore, it is also possible to assume that the dehydrogenation treatment was effective since the hydrogen embrittlement effect did not occur” with “Furthermore, it is also possible to assume that the dehydrogenation treatment was effective since the fatigue life was similar for both coated and uncoated specimens”. Besides, a remark that associates the brittle crack propagation to the high yield stress value and low ductility of the material was added.

If you work on the above remarks, your paper is OK for publication

Thank you again. Your comments were very important to improve the paper. All the above remarks were addressed.

Attached is a report regarding English review.

Best Regards,

Jefferson
